# Multiplex *in situ* hybridization within a single transcript: RNAscope reveals dystrophin mRNA dynamics

John C. W. Hildyard[1]*, Faye Rawson[1], Dominic J. Wells[2], Richard J. Piercy[1]

1 Comparative Neuromuscular Diseases Laboratory, Department of Clinical Science and Services, Royal Veterinary College, London, United Kingdom, 2 Department of Comparative Biomedical Sciences, Royal Veterinary College, London, United Kingdom

* jhildyard@rvc.ac.uk

**Data Availability Statement:** Macros used for image analysis are available at the figshare repository: 10.6084/m9.figshare.9764930

## Abstract

Dystrophin plays a vital role in maintaining muscle health, yet low mRNA expression, lengthy transcription time and the limitations of traditional *in-situ* hybridization (ISH) methodologies mean that the dynamics of dystrophin transcription remain poorly understood. RNAscope is highly sensitive ISH method that can be multiplexed, allowing detection of individual transcript molecules at sub-cellular resolution, with different target mRNAs assigned to distinct fluorophores. We instead multiplex within a single transcript, using probes targeted to the 5' and 3' regions of muscle dystrophin mRNA. Our approach shows this method can reveal transcriptional dynamics in health and disease, resolving both nascent myonuclear transcripts and exported mature mRNAs in quantitative fashion (with the latter absent in dystrophic muscle, yet restored following therapeutic intervention). We show that even in healthy muscle, immature dystrophin mRNA predominates (60–80% of total), with the surprising implication that the half-life of a mature transcript is markedly shorter than the time invested in transcription: at the transcript level, supply may exceed demand. Our findings provide unique spatiotemporal insight into the behaviour of this long transcript (with implications for therapeutic approaches), and further suggest this modified multiplex ISH approach is well-suited to long genes, offering a highly tractable means to reveal complex transcriptional dynamics.

## Introduction

The dystrophin gene is the largest in the genome: at approximately 2.4Mbp in length this single gene accounts for almost 0.1% of human haploid DNA and occupies fully 1.5% of the X chromosome where it resides. Transcription of a single full-length 79-exon dystrophin mRNA is estimated to take 16 hours, with translation giving rise to dp427 dystrophin (a protein product of 427kDa). This lengthy gene also displays unconventional transcriptional behaviour: the dp427 transcript can arise from three separate promoters controlling cortex, muscle and Purkinje cell expression, producing dp427c, dp427m and dp427p respectively [1, 2]. Each promoter confers a unique first exon but the other 78 are shared, resulting in dp427 proteins

**Funding:** This work was funded by the Wellcome Trust [Grant number 101550], awarded to RJP. The funders had no role in study design, data collection and analysis, decision to publish, or preparation of the manuscript.

**Competing interests:** The authors have declared that no competing interests exist.

**Abbreviations:** ISH, In situ hybridization; DMD, Duchenne muscular dystrophy; BMD, Becker muscular dystrophy; NMD, Nonsense-mediated decay; DAGC, Dystrophin-associated glycoprotein complex; nNOS, neuronal nitric oxide synthase; PPMO, PIP6a peptide-conjugated antisense morpholino oligonucleotide; FITC, Fluorescein isothiocyanate; Cy3/Cy5, Cyanine 3/5; SDH, Succinate dehydrogenase; PTC, Premature termination codon.

essentially identical in sequence, differing only in the first 3–11 amino acids (of a total of almost 3700). The dystrophin gene also carries several internal promoters producing shorter transcripts, generating multiple N-terminally truncated protein isoforms (similarly named by molecular weight, thus dp260 [3], dp140 [4], dp116 [5], and dp71 [6]): see Fig 1A. These isoforms show clear tissue-specific expression patterns and likely play unique cellular roles, but most studies to date have focussed on dp427m, the full-length muscle isoform: mutations almost anywhere in the dystrophin gene affect this isoform, and this isoform is critical to human health.

Dp427m is the only dystrophin isoform canonically expressed in skeletal muscle [7], and the long dp427 protein it encodes contains three principal domains: an N-terminal actin-binding region, a C-terminal dystroglycan-binding region, and a lengthy 'rod' domain of 24 spectrin-like repeats linking the two (Fig 1B) [8]. This protein localizes to the sarcolemma (Fig 1C) where it forms a key component of the dystrophin-associated glycoprotein complex (DAGC), co-localizing proteins that aid muscle function such as neuronal nitric oxide synthase (nNOS) [9] and acting as a physical link between the intracellular actin cytoskeleton and the extracellular matrix environment [10]. Insufficient or absent muscle dp427 (Fig 1D) results in weakness of the muscle sarcolemma, leaving muscle fibres vulnerable to contraction-induced injury, leading to the incurable and fatal muscle-wasting disease, Duchenne muscular dystrophy (DMD). This X-linked disease, affecting 1 in 5000 new born boys [11, 12], is characterized by cycles of muscle degeneration and compensatory regeneration, with concomitant persistent inflammation and progressive loss of muscle tissue to fat and fibrotic scarring: boys become wheelchair-bound between the age of 8 and 12, and death typically occurs in the 20s-30s via either cardiac or respiratory failure [13]. Mutations that cause premature termination of dp427 result in DMD, however mutations that cause in-frame internal truncation of dp427 typically lead to the milder condition, Becker muscular dystrophy (BMD), suggesting that while N and C termini are critical for function, the rod domain is largely dispensable: indeed, use of antisense oligonucleotides to mask splicing sites in the dp427 pre-mRNA leads to omission of targeted exons in DMD-causing transcripts and consequent restoration of reading frame, offering a means to change DMD to a BMD phenotype. These 'exon-skipping' approaches represent a promising therapeutic approach.

Dystrophin remains a challenging target of investigation: dp427 represents only a tiny fraction of total muscle protein (~0.002% [14]), and levels of transcripts within skeletal muscle are

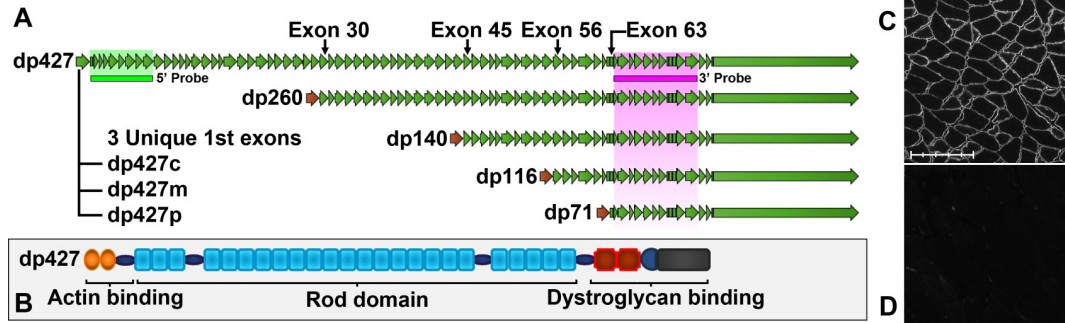

**Fig 1. Dystrophin mRNA isoforms and full-length protein.** (A) Cortical, muscle and Purkinje isoforms of full length dystrophin (dp427) have unique first exons but share all 78 downstream exons. Dp260 shares all sequence from exon 30, dp140 from exon 45, dp116 from exon 56 and dp71 from exon 63. Target sequence regions of the 5' and 3' probes used in this study are shown (green and magenta bars): 5' probe recognizes dp427 only (c, m and p), while 3' probe will recognize all dystrophin isoforms. (B) dp427 protein has three principal domains: N-terminal actin binding, spectrin repeat rod domain, and C-terminal dystroglycan binding. Skeletal muscle dystrophin protein localizes to the sarcolemma in healthy mouse quadriceps muscle (C) but is absent in dystrophic mdx quadriceps muscle (D). Scale bar: 200μm (subdivisions: 40μm).

consequently similarly modest [15, 16]. Immunohistochemistry with specific antibodies allows the subcellular localization of dystrophin protein to be determined with high accuracy, but similar studies at the mRNA level are more challenging. *In-situ* hybridization (ISH) with biotinylated [17] or radiolabelled [18] probes suggests a sarcolemmal localization of the dp427 transcript, but further details are confounded by the innate limitations of this technique and the low transcript abundance. Most studies of dp427 mRNA dynamics have therefore typically measured dystrophin expression in whole muscle extracts, necessarily discarding spatial information. As muscle tissue is host to multiple cell types [19], such approaches also introduce unavoidable noise (a problem exacerbated in dystrophic muscle where infiltrating inflammatory cells, fibroblasts, adipocytes and differentiating myoblasts further diversify the cellular milieu [20]). Despite these limitations, pioneering work by Tennyson *et al* [21] determined estimates for dp427 transcription time of the order of 16 hours, and showed that splicing occurred co-transcriptionally (work subsequently supported by others investigating transcription of long genes [22] where co-transcriptional splicing and elongation rates of 40–60 bases per second were reported). Further investigations [16] suggested that in human muscle dp427 5' sequence was present in excess of 3' sequence (i.e. nascent transcripts exceeded mature), suggesting the half-life of mature dp427 might be modest, but at the time such work necessitated transcript estimation via somewhat crude RT-PCR autoradiography. An elegant recent study by Gazzoli *et al* revealed that co-transcriptional splicing of dp427 occurs non-sequentially, and that long introns in particular (several are >100kb) are spliced in multiple steps [23], however most studies of dp427 focus on disease- or therapeutic-relevant metrics such as the efficiency of exon-skipping (percentage of skipped transcripts) rather than more general properties of the mRNA itself. Recent technological advances such as RNAscope [24] have led to a resurgence in ISH-based investigations, suggesting that detailed study of muscle dp427 mRNA at the histological level might now be practical and (given the large size and low abundance of the transcript, and the unique multinucleate nature of myofibres) also highly nuanced.

RNAscope ISH employs proprietary 'ZZ' oligonucleotide probe pairs: each probe carries ~25 bases of target-complementary sequence linked to one half of a preamplifier-binding motif. The preamplifier carries multiple amplifier-binding motifs, and the amplifiers in turn are able to bind peroxidase enzymes in very high local concentrations. Use of 20 such ZZ pairs in series allows targeting of ~1000 bases of target sequence (rendering this method highly specific), and use of the amplification strategy with peroxidase-activated tyramide dyes affords very high sensitivity, allowing detection of individual mRNA transcripts with sub-cellular resolution (Fig 2A–2E). Crucially, this technique can also be multiplexed: by use of probe-specific amplifiers and sequential labelling, multiple mRNA species can be resolved, each with a different fluorophore (Fig 2F). We reasoned that this approach might be further refined: at 14kb, a full-length dystrophin transcript should permit multiple 20ZZ probe sets to bind simultaneously (a hypothesis supported by Smith *et al*, using conventional fluorescent probes [25] to resolve nascent transcriptional loci within cultured myonuclei). We have previously confirmed the validity of this approach, using a triplex strategy (targeting 5', middle and 3' sequence of the dp427 transcript) to reveal expression of different dystrophin isoforms in the developing embryo [26]: dp427 transcripts in nascent muscle were labelled with all three probes, dp140 transcripts in the developing brain were labelled with only middle and 3', while cell types known to be rich in the short dp71 isoform (such as epithelia) labelled with the 3' probe alone.

Here we study mature skeletal muscle, a tissue known to express only full-length muscle dystrophin (dp427m). Given the ~16 hour transcription time of this long mRNA, use of probes to 5' and 3' sequence of dp427 (see Fig 1A) offers an elegant means to label transcripts in a temporal fashion, in a manner similar to that first used by Garcia *et al* [27], but without

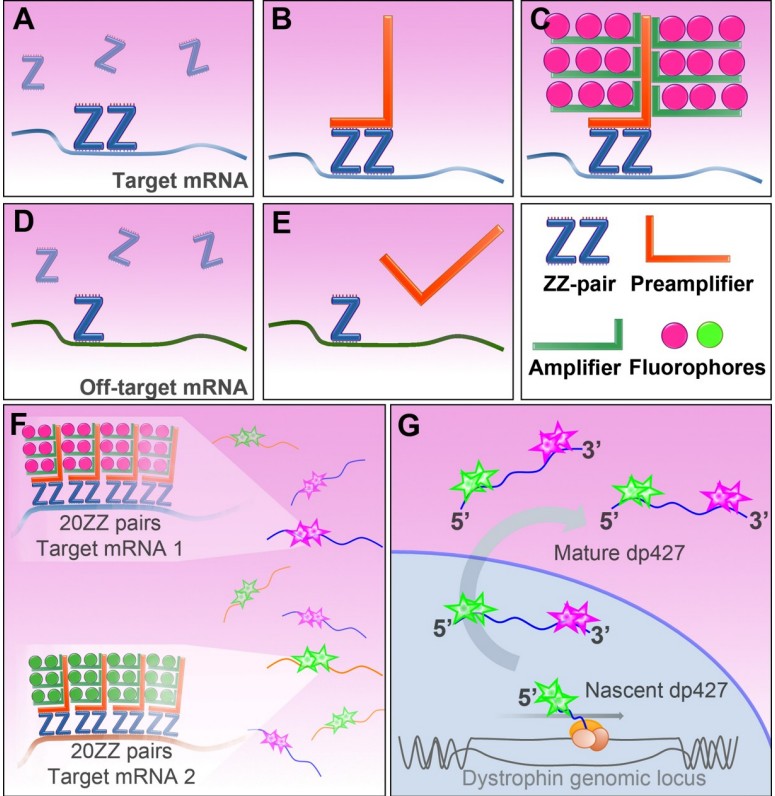

**Fig 2. RNAscope ISH method.** (A) RNAscope ZZ probe pairs bind 30–50 bases of target sequence. Adjacent probes create a target motif for preamplifier molecules (B), which bind amplifiers, which bind probe-specific enzyme to deposit high local concentrations of fluorophore (C). Off-target effects are minimized as both elements of the ZZ pair must bind in close proximity. A single Z probe is insufficient to bind preamplifier (E). Probe sets of 20 sequential ZZ pairs (F) can be used to ensure high specificity, and sequential probe-specific dye-labelling allows different fluorophores to label different mRNA target molecules. (G) dp427 can be targeted with multiple probes: sets targeting 5' (exons 2–10) and 3' (exons 64–75) regions of dp427 permit distinction of nascent (5' only) and mature (5' and 3') transcripts.

recourse to complex transgenics. This approach should allow detection of individual mature and nascent dystrophin mRNA molecules within skeletal muscle: the majority of nascent transcripts would bind 5' only, while mature would bind both (Fig 2G). We have used these probes in healthy wild-type (WT) and dystrophic (*mdx*) mouse muscle, and in *mdx* muscle rescued with exon-skipping agents, to gain insights into the localization and dynamics of this critical transcript in health and disease. Our work supports the findings of previous investigations while adding spatiotemporal refinements unachievable by other means, offering a powerful and versatile new tool not only for the study of dystrophin, but for other transcripts produced from long genes.

## Methods

### Probe design

20ZZ RNAscope probes (ACDBio) were designed to mouse dystrophin sequence (accession number NM_007868.6). The catalogue probe (Mm-Dmd, Cat. No. 452801) recognizes residues 320–1295 (exons 2 to 10) of the full-length (dp427) dystrophin transcript. This probe (henceforth: *5' probe*) recognizes full-length dp427 only, but will bind both nascent and mature

dp427 transcripts. A further custom probe (Mm-Dmd-O1, henceforth: *3' probe*) was designed to residues 9581–10846 (exons 64–75). This 3' probe hybridizes to sequence emerging late in transcription, favouring mature transcripts. Given shared sequence, 3' probe will recognize essentially all dystrophin isoforms (dp427, 260, 140, 116, 71 –see Fig 1A).

Positive control probes to POLR2A (NM_009089.2, residues 2802–3678), PPIB (NM_011149.2, residues 98–856) and UBC (NM_019639.4, residues 36–860) were used to confirm preservation of sample RNA, while negative control probes to bacterial DapB (EF191515, residues 414–862) were used to establish non-specific labelling.

### Ethics statement

A total of ten male mice were used in this study (6 *mdx* and 4 wild-type C57BL/6J). All mice were bred under UK Home Office Project Licence PPL 70/7777 (holder Professor Dominic Wells), approved by the Royal Veterinary College Animal Welfare and Ethical Review Body. Mice were held in open top cages in a minimal disease unit at an average 21˚C in a 12 hours light/ 12 hours dark light cycle with food and water provided ad-lib. Seven 40-week old male mice (3 *mdx*, 4 wild-type–referred to within the text as *mdx* 1–3 and WT 1–4) were collected specifically for this study, while a further three 32-week old male *mdx* mice (untreated–*mdx* 4 within the text-, or treated intravenously with PIP6a peptide-conjugated antisense morpholino oligonucleotide (PPMO) at 6mg.kg$^{-1}$ and 12.5mg.kg$^{-1}$) were archive samples taken from two separate studies ([28] and manuscript in preparation): briefly, mice were treated via tail vein injection with the PPMO starting at 12 weeks of age, continuing for a total of 10 injections each separated by two weeks with collection two weeks after the last dose. All animals were killed by cervical dislocation, with tissues harvested rapidly post mortem. Quadriceps cDNA samples used for qPCR analysis (9 *mdx*, 9 WT) were archive samples from our collection and were not prepared specifically for this study.

### Sample preparation and cryosectioning

Quadriceps muscles used for this study were mounted longitudinally (a configuration that generates longitudinal and transverse regions within a single section) in cryoMbed mounting medium (Bright instruments Ltd) and flash-frozen in liquid $N_2$-cooled isopentane. Freezing in an essentially relaxed configuration rather than pinned out to L0 reduces fibre hypercontraction immediately post-sectioning (see below). Muscle tissues were sectioned at -25˚C to 8μm thickness using an OTF5000 cryostat (Bright) and mounted on glass slides (SuperFrost, VWR). Preparation of longitudinal sections from frozen mouse muscle is non-trivial: transient melting during transfer of section to slide typically results in considerable fibre hypercontraction as contractile proteins respond to abundant free calcium ions. RNAscope ISH involves multiple detergent washes and incubations at 40˚C, necessitating the enhanced tissue affinity of SuperFrost slides, but this slide format is incompatible with EDTA-coating methods of Pearson and Sabarra [29]. It was determined that freezing muscles in a relaxed state and sectioning at 8 micron thickness generates acceptable longitudinal sections without the requirement for EDTA: at this thickness, shear forces generated by contractile proteins in the presence of free Ca2+ appear largely insufficient to overcome the rapid adhesion to SuperFrost slides. Serial sections were collected and slides were air-dried at -20˚C for 1 hour before storage at -80˚C until use.

### Staining

**Immunofluorescence.** Slides were allowed to equilibrate to room temperature, blocked for 1hr using 5% milk powder (Marvel) in PBS + 0.05% tween (PBS-T), and then co-

immunolabelled with rabbit polyclonal anti-dystrophin (ab15277, Abcam, 1:800) and rat monoclonal anti-perlecan (clone A7L6, Thermofisher, 1:2000), in PBS-T for 1hr. Secondary labelling used Alexa-fluor conjugated antibodies (anti-rabbit 488 and anti-rat 594 respectively, Thermofisher, both 1:1000, 1hr). Nuclei were stained using Hoechst 33342 (Thermofisher, 1:2000, 10 minutes), and slides were then washed and mounted using Hydromount (National Diagnostics).

**Succinate dehydrogenase.** Slides were allowed to equilibrate as above, then incubated with freshly-prepared SDH staining solution (125mM sodium succinate, 1.5mM nitroblue tetrazolium, 60mM Tris HCL, pH 7.0) at 37˚C for 1 hour, fixed in formal calcium for 15mins at room temperature, then washed and mounted in Hydromount.

## RNAscope slide preparation

The combination of protease digestion and detergent-rich wash buffer used by RNAscope is incompatible with fresh-frozen muscle prepared according to standard protocols: while connective tissue/extracellular matrix elements (and most nuclei) were readily retained throughout the protocol, myofibrillar content was rapidly lost leaving empty myofibre 'ghosts' (see S1 Fig). Preparation of frozen muscle thus required extended fixation and an additional baking step (both indicated with an asterisk): slides were removed from -80˚C storage and placed immediately into cold (4˚C) 10% neutral-buffered formalin, then incubated at 4˚C for 1 hour*. Slides were then dehydrated in graded ethanols (50%, 70%, 100% x2, 5mins in each, room temperature) and left in 100% ethanol at -20˚C overnight, then air-dried and baked at 37˚C for 1 hour*. Sections were ringed using hydrophobic barrier pen (Immedge, Vector Labs) and then treated with RNAscope hydrogen peroxide (ACDbio) for 15mins at room temperature to quench endogenous peroxidase activity. After washing twice in PBS, slides were protease-treated (RNAscope Protease IV, ACDbio) for 30mins at room temperature and washed a further two times in PBS before use in RNAscope multiplex assay (see below).

## RNAscope multiplex assay

Multiplex assays were performed according to the manufacturer's protocols. All incubations were at 40˚C and used a humidity control chamber (HybEZ oven, ACDbio). Probe mixes used were as follows:

- RNAscope 3-plex positive control probe set (320881): POLR2A, PPIB and UBC (low, moderate and high expression targets supplied as C1, C2 and C3 probes, respectively)

- RNAscope 3-plex negative control probe set (320871): Bacterial DapB (in C1, C2 and C3 probe sets)

- RNAscope mouse dystrophin probe set: Mm-Dmd (452801) and Mm-Dmd-O1 (custom probe) (C1 and C2 probes, targeting 5' and 3' sequence respectively).

Tyramide dye fluorophores (FITC, Cy3, Cy5: TSA plus, Perkin Elmer NEL760001KT) were used diluted appropriately in RNAscope TSA dilution buffer. Nuclei were labelled with RNAscope DAPI (ACDbio) for 30 seconds, or Hoechst (1/2000 in wash buffer for 10mins followed by two washes). Slides were mounted in Prolong Gold Antifade mounting medium (Thermofisher).

Positive and negative control slides were typically given the following fluorophore assignments: C1 –TSA-FITC; C2 –TSA-Cy3; C3 –TSA-Cy5. Due to weak FITC fluorescence and strong signal intensity of the 5' dystrophin probe (as compared with the 3' probe), many combinations of fluorophore posed either poor probe comparability or signal bleed-through

(particularly those using TX2 filter cubes). Our preferred final fluorophore combination for dystrophin was therefore TSA-Cy3 and TSA-Cy5 (for C1 5' and C2 3', or switched as indicated). Used with the N3 and Y5 filter cubes, this combination of fluorophores had no signal overlap and allowed selection of modest exposure times without bleed-through. TSA dyes were used at the following dilutions: TSA-FITC, 1/500; TSA-Cy3, 1/1500; TSA-Cy5, 1/750.

## Imaging

Individual images were collected using either a DM4000B or DMRA2 upright fluorescence microscope with samples illuminated using an EBQ100 light source and A4, L5, N3 and Y5* filter cubes (Leica Microsystems) and an AxioCam MRm monochrome camera controlled through Axiovision software version 4.8.2 (Carl Zeiss Ltd). For all analysis, images were captured via 20x objectives (20x HC PL FLUOTAR PH2, NA = 0.5) which readily enabled identification of discrete foci corresponding to individual transcripts, while retaining sufficient axial resolution to allow all elements of the tissue section to remain well-focussed. Multiple images (9–30) were collected per muscle section, selected randomly (avoiding section tears and tendinous regions). For particle counts, exposure times were calculated to allow clear detection of small foci in all channels (typically 600-1500msec). For fluorescence intensity (3–5 images per muscle section), multiple exposure times were collected (100-1000msec) to minimize risk of signal saturation. For whole section alignments, images were collected at 5x and stitched using the pairwise-stitching algorithm of Preibisch *et al* [30]. After analysis, images shown in figures were adjusted for clarity using the window/level tool (imageJ).

## Image analysis

Collected images (as tiff format files) were analysed using the Fiji distribution of ImageJ, using automated macros (all requisite.ijm files can be found at the figshare repository at the following DOI: 10.6084/m9.figshare.9764930). In brief, images were separated into Cy5, Cy3 and DAPI channels: for counts of nuclei, nuclear stain was thresholded automatically using either Huang or Yen's algorithm [31] and given a single binary watershed pass to separate adjacent nuclei. Cy3 and Cy5 channels were thresholded to eliminate background/non-specific staining and foci were counted using the 'Analyze Particles' tool. Fluorescence microscopy cannot reveal the true size of fluorescent particles (instead resolving a point-spread function [32]) but apparent size allows discriminatory analysis: apparent particle areas were converted to square microns using the appropriate multiplier for the magnification and image resolution. For nuclear/extranuclear particle analysis, the DAPI channel was thresholded as above and applied as a mask to the Cy3/Cy5 channels (eliminating nuclear-localized signal) or applied as an inverted mask (eliminating non-nuclear signal) prior to thresholding and particle counting. Nearest neighbour analysis used extranuclear foci only, with analysis then repeated using the same images with the 3' channel rotated by 90 degrees (distributions for random points were obtained from the average of ~1000 virtual fields of randomly-generated points of specified total number). For 5' probe fluorescence intensity analysis (due to stark differences in 5' signal intensity) 3–5 images per animal were collected at multiple exposure times (see above): regions of interest (sharply-defined small foci, broad nuclear probe signals, and background regions; 30–50 per image, 10–15 for background) were determined from the highest exposure, then quantified at lower exposure times using only non-saturated signal.

## Calculations

Mean values of transcripts per field were used to calculate total transcript numbers via the following calculations: 1388x1040 pixel imaging field at 20x (~2 pixels per micron) = 360880μm$^2$

at a section thickness of 8um, thus 2889040μm$^3$ (2.9x10$^{-3}$mm$^3$). At a muscle density of 1.06g. ml$^{-1}$ [33], a single imaging field thus represents ~3μg of tissue. Typical RNA yields per mg of muscle tissue are 0.25–0.5μg, thus a single 20x imaging field can be considered to contain 0.75–1.5ng of total RNA.

**Calculation of transcript half-lives.** Half-life calculations used the method provided by Tennyson and colleagues [16]. At steady-state, the ratio of 5' sequence (nascent and mature dp427) to 3' sequence (mature only) represents the balance of transcription time to mean lifetime.

$$\frac{5'}{3'} = \frac{nascent + mature}{mature} = \frac{T_{transcrip} + T_{lifetime}}{T_{lifetime}}$$

The half-life of a given species is related to the mean lifetime by the equation $T_{lifetime} = T_{1/2}/0.693$.

Our 5' probe sequence ends at exon 10 and the 3' at exon 74 (1550kbp between the two) while our qPCR regions (see below) were designed to the exon 1:2 and exon 62:63 boundaries (1950kbp); assuming 16 hour constant-rate transcription for full-length dp427 [21] this gives an approximate transcription time between 5' and 3' of ~11 hours for RNAscope probe regions and 13.5 hours for qPCR amplicon regions.

## RNA isolation and qPCR

cDNAs used for qPCR (male quadriceps muscle, 9 WT, 9 *mdx*) were archive samples from our collection. Isolation and preparation methods were as described previously [34, 35]: RNA was isolated from muscle powder (homogenized under liquid nitrogen) using TRIzol reagent (Invitrogen) with inclusion of an additional chloroform extraction (1:1) after phase separation. RNA was assessed to determine yield and purity (nanodrop ND1000) and used to prepare cDNA via RTnanoscript2 (Primerdesign). Triplicate qPCRs used 10μl volumes (~8ng cDNA per well assuming 1:1 conversion) in a CFX384 lightcycler using PrecisionPLUS SYBR green qPCR mastermix (Primerdesign), with a melt curve included as standard. Primers to *ACTB*, *RPL13a*, *CSNK2A2* and *AP3D1* (reference genes determined previously [34]) were taken from the geNorm and geNorm PLUS kits (Primerdesign). Primers to dystrophin dp427 and dp71 were designed using primer3 software (http://primer3.ut.ee/) and all span one or more introns (exons 62 and 63 are very short -61 and 62bp respectively- thus the reverse primer for this amplicon lies in exon 64, consequently spanning intron 62 and 63). All primer sets gave PCR efficiencies of 90–105% and all produced a single amplicon. Absolute quantification of transcript numbers was performed via standard curve (dilution series of purified, quantified PCR products from 10^7 molecules per well down to 10^-2), allowing accurate measurement of transcript numbers down to ~10–100 molecules.well$^{-1}$ (typical values from muscle cDNAs were 500–50000 transcripts.well$^{-1}$).

Dp427m Exon1 Fwd 5'-TCTCATCGTACCTAAGCCTCC-3'
Dp427 Exon2 Rev 5'-GAGGCGTTTTCCATCCTGC-3'
Dp427 exon44 Fwd 5'-TGGCTGAATGAAGTTGAACAGT-3'
Dp427 exon45 Rev 5'-CCGCAGACTCAAGCTTCCTA-3'
Dp427 exon62 Fwd 5'-AGCCATCTCACCAAACAAAGT-3'
Dp427 exon64 Rev 5'-ACGCGGAGAACCTGACATTA-3'
Dp71 exon1 Fwd 5'-GTGAAACCCTTACAACCATGAG-3'
Dp71 exon2 Rev 5'-CTTCTGGAGCCTTCTGAGC-3'

## Statistical analysis

Statistical analyses of nuclear counts and fluorescent foci counts used two-tailed Mann-Whitney U tests, while correlation analysis (5' vs 3' foci, total foci vs SDH or nuclear count) used

Pearson r or Spearman's rho as appropriate (indicated). Significance was set at $P < 0.05$. To confirm adequate sample size for fluorescence intensity analysis, values for small 5' foci (means, standard deviations) obtained from three WT mice were analysed using the post-hoc methods of Snedecor and Cochrane [36]. 70–80 foci are necessary to determine the mean with precision of ±10% at 95% confidence: numbers of measured foci (84–112 per individual) are sufficient in all cases. All statistical analyses were conducted using Graphpad Prism 7 and Microsoft Excel.

## Results

### Multiplex ISH to 5' and 3' regions of dp427 allows detection of nascent and mature transcripts

RNAscope *in-situ* labelling allows resolution of individual mRNA molecules, particularly when used with low-abundance transcripts [24], and the expected staining pattern is that of small, punctate foci with an apparent diameter of 1–1.5μm distributed throughout the cell interior. Muscle labelled with positive control probes (to Polr2a, Ppib and UBC: very low, low and high-expression in skeletal muscle, respectively [37]) behaved consistently with these expectations (S2A and S2B Fig). Negative control probes (bacterial DapB in all channels) produced no foci (S2C and S2D Fig), though diffuse, weakly-stained signal was found within small patches of cells in all dystrophic tissues, with all fluorophores. These were also observed in dystrophic positive control slides, but not in healthy muscle (see S2 Fig, arrowheads) and most likely represent aberrant tyramide dye deposition within infiltrating macrophages (these cells are common to dystrophic tissue, and are rich in endogenous peroxidases that might not be fully quenched). In contrast, muscle labelled with RNAscope probes to dystrophin 5' and 3' sequence revealed a striking and highly probe-specific distribution of signal. In healthy muscle (Fig 3) 5' probe signal fell into two separable categories: the expected population of small, punctate foci distributed along the entire body of the myofiber (with the highest concentrations found closest to the sarcolemma), but also a second, less-numerous population of broad, heterogeneous and high-intensity signals localized exclusively to specific nuclei. The intensity of this latter population was such that these large 5' foci were readily detectable even at low magnification. 3' probe signal displayed essentially only the former punctate signal, both within nuclei and without; notably, these foci were typically found adjacent to 5' foci. These distributions were consistent throughout the entire muscle section, both in transverse and longitudinal fibres, and reciprocal patterns were observed if the fluorophores chosen were exchanged (i.e. signal intensity and localization is a property of the probes alone: S3 Fig).

The genomic sequence of the dp427 region targeted by our 5' probe (exons 2–10) begins ~190kb downstream from dp427m exon 1 and spans ~525kbp: assuming a constant, sequence-independent transcription rate of ~40bases.sec$^{-1}$ with concurrent splicing (suggested by the 16-hour transcription time for a 2.3Mb gene, and supported by studies of other long genes [21, 22]), the first nucleotides of this sequence should emerge ~1.3 hours after initiation, with the full 20ZZ probe-binding region being completed ~3.5 hours later and persisting within the nucleus for a further ~11 hours. In contrast, 3' probe sequence (exons 64–75) begins 2.1 megabases downstream of exon 1 and concludes a mere ~88kb later, only 28kb from the transcription termination site. Under the assumptions above, full 3' probe sequence emerges less than 15 minutes before transcript completion, and subsequent nuclear export of completed mRNA is likely to be rapid [38] (especially compared to dp427 transcription time).

Consistent with this, nuclear staining for 5' sequence (but not 3') demonstrates our approach can detect nascent dp427 transcripts. The intensity of nuclear 5' labelling further implies that multiple nascent transcripts are present within any given nucleus: myonuclear

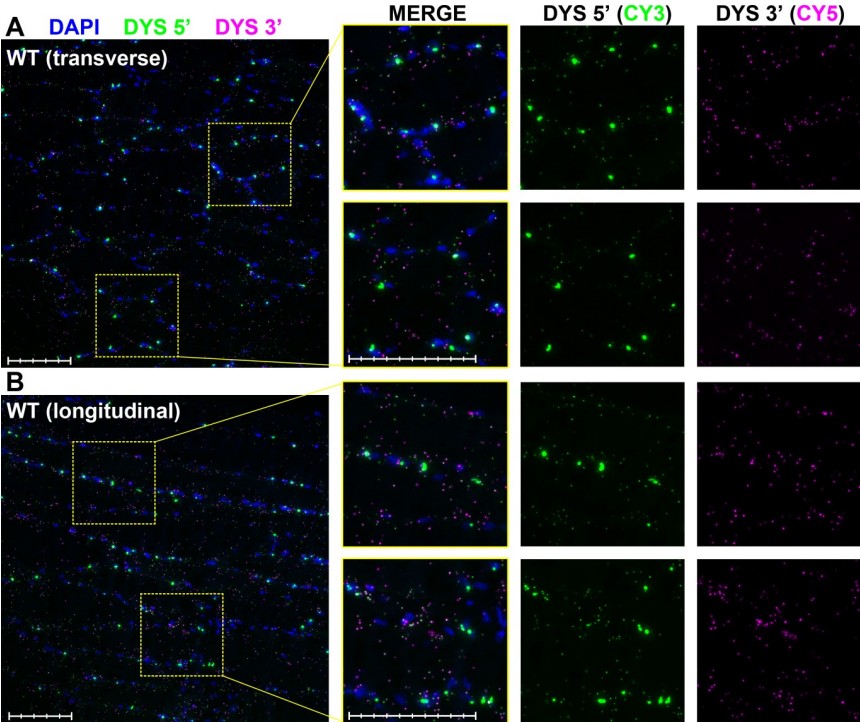

**Fig 3. Dystrophin multiplex ISH in healthy mouse muscle.** RNAscope ISH labelling of dp427 5' and 3' in 40-week old WT quadriceps. Probe to dp427 5' (Cy3: green) resolves both small sarcoplasmic foci and large nuclear foci, while 3' probe (Cy5: magenta) shows small foci only. Foci are readily detected both in transverse (A) and longitudinal (B) section. Scale bars: 100μm (main panel subdivisions: 20μm; magnified panel subdivisions: 10μm).

commitment to dp427 expression might be concerted. Smaller foci (both 5' and 3') found outside nuclei should thus correspond to dual-labelling of mature, exported dp427 transcripts. Supporting this, labelling of dystrophic *mdx* muscle with our probes (Fig 4) revealed key differences to healthy muscle: prominent 5' nuclear foci were readily detected, including within centrally-located nuclei characteristic of regenerated *mdx* muscle (see left side of Fig 4B), but smaller punctate foci from both probe sets appeared greatly reduced in number, and were almost exclusively found within (or adjacent to) nuclei. Additionally, dystrophic muscle also exhibited rare nuclei associated with multiple 3' foci alone (Fig 4B, Arrows), a labelling pattern consistent with expression of the short dystrophin isoform dp71 (as we have shown previously [26]).

The well-defined ISH foci produced by low-abundance transcripts are amenable to quantitative image analysis, allowing measurement of both particle number and apparent area. Confirming our observations above, healthy muscle particle counts were significantly higher than those for dystrophic muscle for both 5' and 3' probes (Fig 5A and Table 1): 600–1400 small foci (of either probe) were found in every WT muscle image, compared with only 100–300 in *mdx*. Per-image variation was however notable even within a given muscle section (S4A Fig) and we speculated this might correlate with fibre type or myofibre nuclear number. Alignment of images with SDH-stained serial sections revealed a correlation between particle count and SDH activity (i.e. oxidative fibres might express higher levels of dp427) however these regions were also host to greater numbers of nuclei, and particle counts correlated better with total nuclei than with SDH (S4B–S4D Fig), suggesting this is the principal source of within-section variation. As normalizing counts to nuclei would bias against dystrophic muscle (infiltrating non-muscle cells mean nuclear counts per field are significantly higher in this tissue: S4E Fig), all analyses used non-normalized, per-image counts.

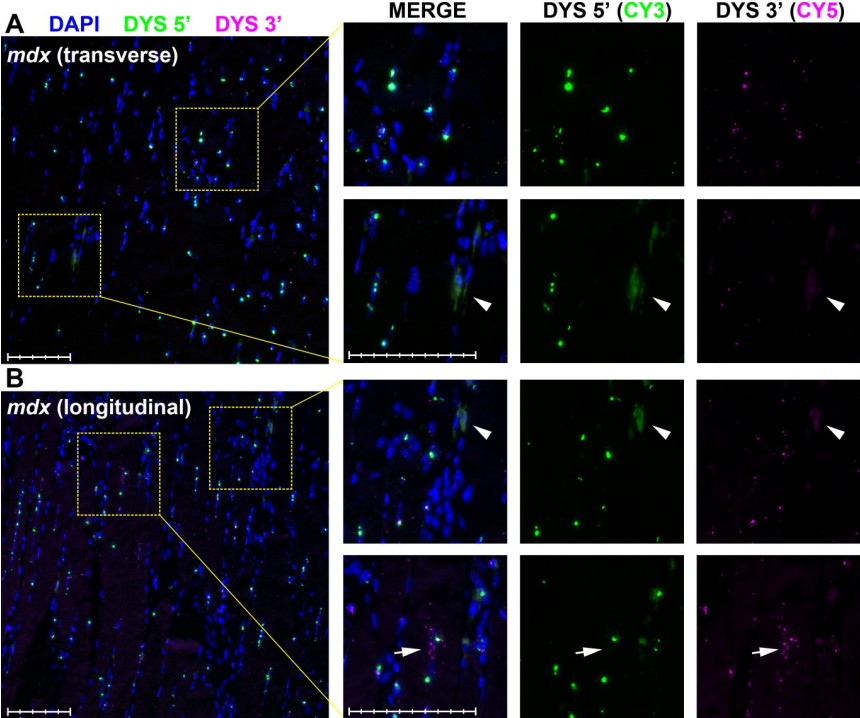

**Fig 4. Dystrophin multiplex ISH in dystrophic mouse muscle.** RNAscope ISH labelling of dp427 5' and 3' in 40-week old *mdx* quadriceps muscle. Probe to dp427 5' (Cy3: green) resolves large nuclear foci but numbers of small foci are greatly reduced. 3' probe (Cy5: magenta) reveals small foci only within or adjacent to nuclei. Nuclear foci are readily detected both in transverse (A) and longitudinal (B) section. Rarely, nuclei with many 3' foci not clearly associated with 5' nuclear foci are observed (Arrows, lower panels). As with positive and negative control probes, apparently cell-restricted regions of non-specific staining are frequently observed (Arrowheads, middle panels), likely corresponding to peroxidase-rich macrophage/neutrophil infiltration. Scale bars: 100μm (main panel subdivisions: 20μm; magnified panel subdivisions: 10μm).

Despite marked differences in total count, healthy and dystrophic muscle exhibited similar partitioning of apparent particle size (Fig 5B): signal from the 3' probe was near-exclusively comprised of foci 1–2μm$^2$ in area while 5' probe signal could instead be segregated into two discrete populations, one of small foci similar to 3' probe, and a second, larger and more heterogeneous population spanning 10–100μm$^2$. We designated particles between 0.5–10μm$^2$ as 'small foci' (95% of this population fell between 1–3μm$^2$) and those 10–100μm$^2$ as 'large foci'. As shown in Fig 5C and Table 1, counts of small foci were markedly lower in dystrophic muscle, while counts of large 5' foci were comparable between genotypes (60–100 per image, accounting for 20–30% of total WT nuclei and, given the greater numbers of nuclei per field, 10–15% of *mdx*). Given the clear nucleus/sarcoplasm delineation within myofibres, we further segregated foci by subcellular location (Fig 5D and 5E), confirming that large 5' foci were indeed restricted to nuclei (consistent with multiple nascent transcripts), while smaller foci of both probes occupied nuclear and sarcoplasmic compartments (consistent with individual completed transcripts). Crucially, counts of small nuclear foci were comparable between WT and *mdx*: only the sarcoplasmic fractions of these small foci were reduced in dystrophic muscle. The dp427 mRNA produced by *mdx* mice carries a premature termination codon (PTC) in exon 23 [39] and is subject to nonsense-mediated decay (NMD) [40], but as this checkpoint occurs after nuclear export [41], pre-export transcripts should remain.

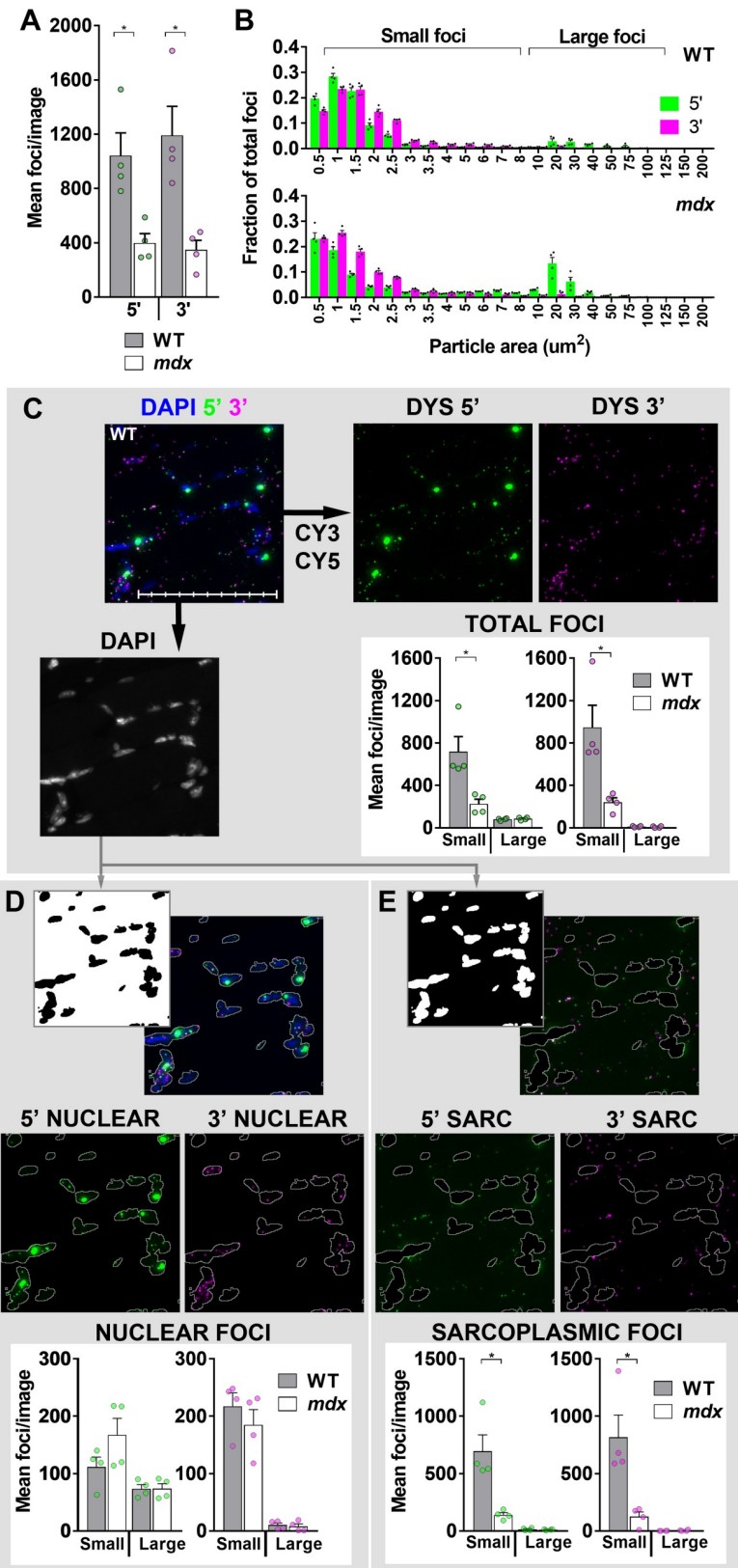

**Fig 5. Particle analysis.** (A) Total particle counts + SEM (N = 4 per genotype) for 5' and 3' probes in healthy (WT) and dystrophic (*mdx*) muscle: values for each animal are shown. (B) Particle size distributions in WT and *mdx* muscle as fraction of total (per image) particle number: brackets used to define small/large foci are shown. (C) Segregation by small/large foci (as indicated) of total 5' and 3' signals, or signals after masking by nuclear stain (DAPI) to give nuclear (D–all sarcoplasmic foci removed) or sarcoplasmic (E–all nuclear foci removed) counts. Note that large foci are found only within nuclei, and only small sarcoplasmic foci (both 5' and 3') are significantly reduced in dystrophic muscle. Representative image shown (same base image in all instances) is taken from a WT sample. Individual mean values per animal are shown as dots (○: green = 5'; magenta = 3') for all counts, or as dots (•) for size distributions. Each mean represents the average of 9–30 images (same panel of images used for all analyses). * = P<0.05, Mann-Whitney U test. Scale bar: 100μm (subdivisions: 10μm).

## 5' and 3' probes co-localize and reveal mature dp427 mRNAs within dystrophin positive revertant fibres

If mature dp427 transcripts can be dual-labelled, 5' and 3' signals should co-localize in healthy muscle, but not dystrophic. WT muscle (Table 1, Fig 5C) exhibited a strong correlation between counts of small 5' and 3' foci: on a per-image basis, Pearson correlations in this fraction ranged from 0.91–0.97 (all P<0.0001), contrasted with 0.41–0.85 (P values from non-significant to <0.0001) for *mdx* muscle (S5A and S5B Fig). As shown in Fig 6A, many sarcoplasmic 5' and 3' foci appear to be in close apposition (arrowheads). We therefore measured 5'/3' co-localization in these foci using a nearest-neighbour approach, determining minimum distance between signals of opposing probes (as compared with the 3' channel rotated, or with equivalent numbers of randomly-distributed points, see Fig 6B). In healthy muscle (Fig 6C, WT) almost 40% of all foci are paired within 1μm, whether 5' to 3' or the reverse, something observed rarely (<1%) with equivalent numbers of randomly-distributed points (indeed, all pairings below 4μm were over-represented in our analysis when compared with random distributions -see S5C Fig). Analysis of the same images following 90° rotation of the 3' channel was instead consistent with random distribution. In *mdx* muscle (with only ~100–200 sarcoplasmic per image), many foci were more than more than 30μm from a focus of the opposite probe: comparable with a random distribution or channel rotation. Sub-micron co-localizations were however still observed, at levels approaching 25–30% of foci in some images (Fig 6C, *mdx*). Again, such close association was not found with random distributions or channel rotation. Closer inspection showed these foci typically were found in small, fibre-restricted, domains with sarcoplasmic 5'/3' counts comparable to healthy muscle.

**Table 1. Mean per-image 5' and 3' particle counts for the eight quadriceps muscles examined.**

| | 5' counts | | | 3' counts | | |
|---|---|---|---|---|---|---|
| | Total | Small | Large | Total | Small | Large |
| WT 1 | 1229 ±231 | 1145 ±221 | 84 ±14 | 1579 ±454 | 1571 ±446 | 8 ±9 |
| WT 2 | 676 ±371 | 586 ±349 | 90 ±26 | 735 ±391 | 718 ±373 | 16 ±23 |
| WT 3 | 641 ±282 | 559 ±265 | 82 ±21 | 806 ±319 | 789 ±302 | 17 ±19 |
| WT 4 | 650 ±160 | 585 ±154 | 64 ±11 | 718 ±181 | 714 ±178 | 5 ±4 |
| *mdx* 1 | 234 ±58 | 152 ±44 | 82 ±23 | 326 ±99 | 324 ±97 | 2 ±3 |
| *mdx* 2 | 386 ±118 | 289 ±101 | 97 ±27 | 290 ±164 | 276 ±151 | 14 ±14 |
| *mdx* 3 | 385 ±76 | 316 ±69 | 69 ±24 | 253 ±69 | 237 ±61 | 15 ±9 |
| *mdx* 4 | 241 ±39 | 146 ±32 | 96 ±17 | 127 ±28 | 126 ±26 | 1 ±2 |

All measured particle counts, expressed as 'total' or subdivided by size (foci 0.5–10μm$^2$ were classed as 'small', those >10μm$^2$ as 'large'). Values shown are means ± standard deviations derived from 9–30 separate 20x images per muscle. Total and 'small' counts were significantly greater in WT muscle than dystrophic, while 'large' counts were not (P<0.05, Mann-Whitney U test). All mean values are shown graphically in Fig 5A and 5C.

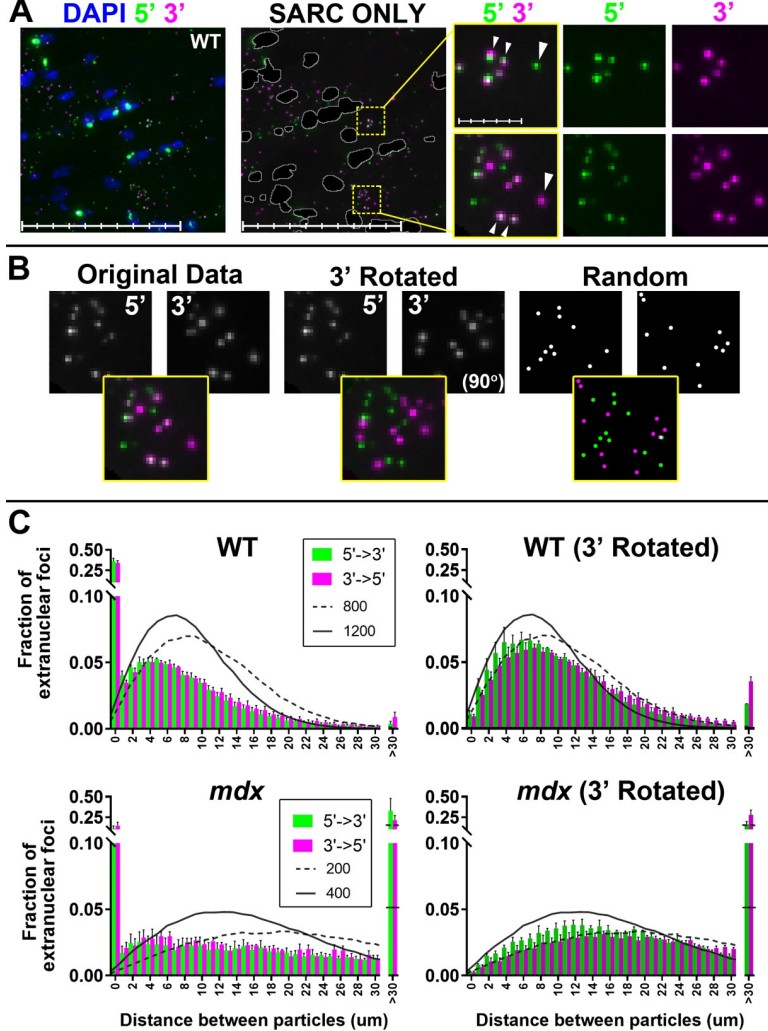

**Fig 6. Co-localization of 5' and 3' signals.** (A) 5' and 3' foci are found in close proximity. Images were masked to restrict analysis to sarcoplasmic signals only. Foci closer than 1μm are indicated (small arrowheads) though isolated foci of either probe were also observed (large arrowheads). Main panel scale bars: 100μm (subdivisions: 20μm); magnified panel scale bars: 10μm (subdivisions: 2 μm). (B) Analysis parameters: distances between foci of 5' and 3' probe channels were examined for raw images, or the same images with the 3' channel rotated by 90 degrees. Representative image shown (same base image in all instances) is taken from a WT sample. A further set of analyses were performed using virtual images with appropriate numbers of randomly distributed points. (C) Distribution of distances between 5' and 3' sarcoplasmic foci. X-axis: distance between a given small 5' sarcoplasmic signal and the nearest 3' (or vice versa) in microns, Y-axis: fraction of total foci paired within that distance for WT (upper panels) and mdx (lower panels) quadriceps muscle. Left hand side: 5'-3' distances are represented as green bars, 3'-5' distances as magenta (bars = means+SEMs, N = 4). ~40% of WT and ~15% of *mdx* foci co-localize within 1 micron while <1% of WT and >30% of *mdx* foci are more than 30μm from a focus of the opposite probe. Right hand side: the same data analysed after 3' channel rotation. Overlaid traces represent expected mean distributions for 800 and 1200 random points, or 200 and 400 random points (as indicated). With 200 or 400 random points, >30μm distances comprise 25% and 5% of all pairings, while for 800 or more, typically <1% of points are paired at such distances.

We reasoned these foci might represent corrected dp427 transcripts expressed within sporadic dystrophin-positive revertant fibres [42], and as shown in Fig 7, these regions align with revertants in serial sections immunolabelled for dystrophin protein. RNAscope images collected only from regions aligning with revertant fibres (Fig 7D) showed higher mean sarcoplasmic 5' counts than images collected randomly, though the increase was not uniform: some

regions of revertant fibres showed no concomitant increase in probe labelling, implying that domains of dp427 mRNA might be more focal than those of dp427 protein.

## Quantification of nascent and mature transcripts shows mature dp427 mRNA is short-lived

The necessity for separate fluorophores precludes direct comparison between 5' and 3' probe fluorescence, but appropriate selection of exposure time (S6 Fig) renders small and large populations of 5' probe foci quantitatively tractable. Fluorescence intensity values of small sarcoplasmic foci are comparable between foci, images and individuals (Fig 8A, small foci): as these

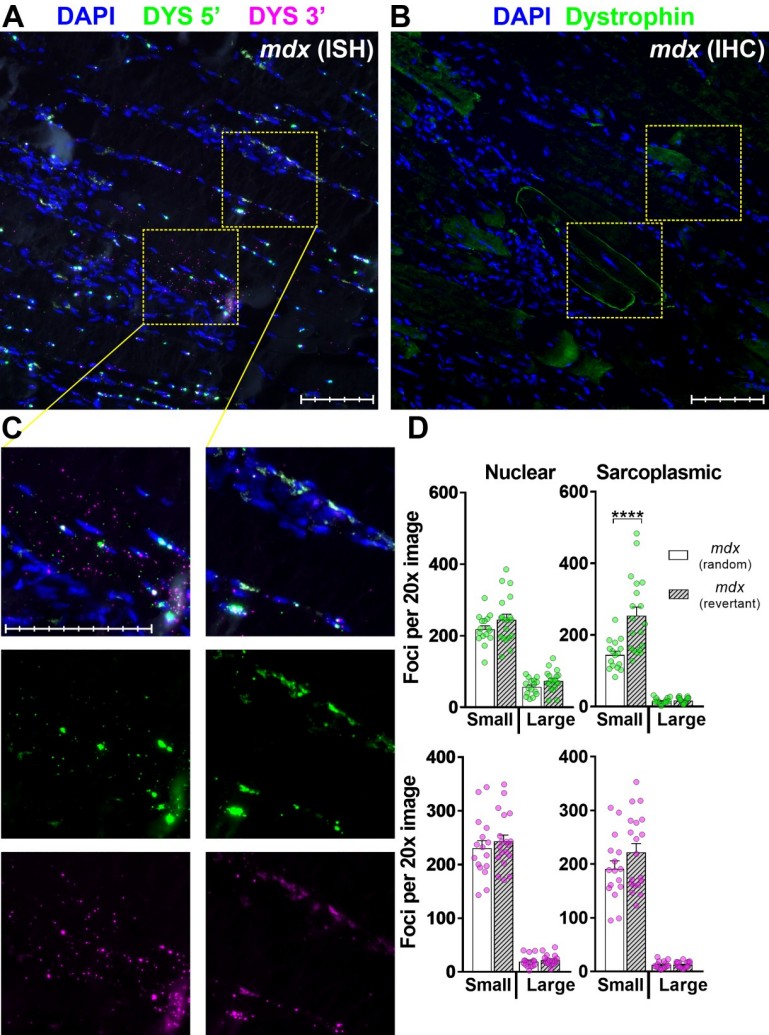

**Fig 7. dp427 expression in revertant fibres.** Aligned serial sections of *mdx* muscle labelled with RNAscope ISH 5'/3' probes (A) and IHC for dystrophin C-terminal antibody (B). RNAscope labelled regions corresponding to dystrophin-positive revertant fibres show large numbers of sarcoplasmic 5' and 3' foci (C, 1st column) while those corresponding to adjacent dystrophin-negative fibres do not (C, 2nd column). Note also presence of rare nuclei labelled with 3' probe alone in surrounding (non-revertant) tissue. (D) nuclear and sarcoplasmic counts of 5' and 3' foci from a single RNAscope labelled dystrophic muscle using images collected at random or only from regions aligned with revertant fibres as indicated: revertant images show increased numbers of small sarcoplasmic 5' foci (Each dot represents a single 20x image, P<0.0001, Mann-Whitney U test). Scale bars: 100μm (main panel subdivisions: 20μm; magnified panel subdivisions: 10μm).

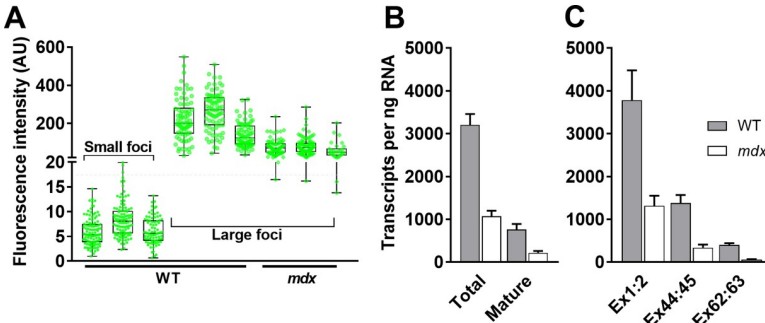

**Fig 8. Nascent and mature dp427 transcripts.** (A) Fluorescence intensity values from representative samplings of well-focussed non-saturated foci from small sarcoplasmic and large nuclear 5' probe populations. Values obtained from three WT and three *mdx* mice (large foci only from *mdx* due to minimal numbers of small foci), using 3 images (60–120 particles in total) per individual. As single transcripts, values for small foci fall over a relatively narrow range and are moreover comparable between individuals (means±SD: WT1 6.0±2.7; WT2 8.4±3.5; WT3 6.1±2.7 AU) and mean values can thus be used to estimate 5' probe content of more heterogeneous large nuclear deposits (typically WT 100–400 AU; mdx 50–150 AU). AU = arbitrary units. (B) Estimated numbers of dp427 transcripts in healthy and dystrophic mice by RNAscope counts: total transcripts (mean estimate of 5' transcripts per nucleus multiplied by number of large nuclear foci, plus all mature foci) vs mature transcripts only, per 20x imaging field (equivalent to ~1ng RNA–see methods); (C) qPCR values: absolute transcript numbers per ng RNA as determined by qPCR to specific dystrophin dp427m exons as indicated, compared against standard curves. Data normalized to the geometric mean of APD3D1, ACTB, CNSK2A2 and RPL13a. All data shown as means + SEM (RNAscope N = 4, qPCR N = 9).

represent single transcripts, comparing the mean single-transcript fluorescence intensity with that of large myonuclear foci (Fig 8A, large foci) allows numbers of nascent nuclear dp427 molecules to be estimated. Our analysis suggests that in healthy muscle most myonuclei contain 20–40 immature transcripts, though expression falls over a considerable range: rare nuclei apparently have only 1–2 transcripts while others appear to contain 60 or more (some higher values may reflect closely-apposed nuclei being counted together). Dystrophic myonuclei exhibit a similar spread of values, but expression as a whole is markedly reduced, with mean values suggesting only 10–12 nascent transcripts per nucleus.

This data, combined with the counts of mature dp427 mRNA molecules above, allows estimation not only of total dystrophin expression, but also of the distribution of such expression between mature mRNA molecules and transcripts in progress (Fig 8B). In dystrophic muscle immature transcripts comprise the bulk of signal as would be expected, but even within healthy muscle, prominent nuclear 5' foci still represent ~10% of the total 5' signals: with each dp427-expressing nucleus host to 20–40 immature transcripts, the implication is that under normal healthy conditions nascent transcripts account for 60–80% of total dp427 mRNA.

We corroborated these estimates by quantitative RT-PCR (qPCR) targeted to specific 5' or 3' regions of the dystrophin gene: using primers spanning the exon:exon junctions at 1–2, 44–45 and 62–63, and quantifying absolute transcript numbers, we show that numbers of detected transcripts do indeed decline as the target sequences approach the 3' terminus, but in dystrophic muscle even 5' signal is reduced, with transcripts containing exons 62–63 being reduced yet further (to levels approaching zero). We also measured dp71 expression, showing this transcript is indeed present at low levels within skeletal muscle (see S8 Fig).

As shown (Fig 8C) qPCR-derived dp427 transcript numbers agreed closely with those obtained via RNAscope both for healthy and dystrophic samples, and confirm that even in healthy muscle the bulk of dp427 is immature. This marked ~3-4-fold bias toward incomplete transcripts, combined with the estimated 16-hour transcription time for a single dp427 mRNA (see methods), suggests that once completed and exported, a mature dystrophin transcript has an average lifespan of only ~3.5 hours, substantially shorter than the time invested in its synthesis.

## Single-transcript multiplex ISH permits quantification of dystrophin induction via exon skipping

Finally, to investigate whether our ISH probes could detect rescued dp427 transcripts, we examined dystrophic muscle treated with an exon-skipping agent. The *mdx* mouse carries a PTC in exon 23 of dp427: this exon lies in-frame with those either side, and can thus be 'skipped' (omitted from the transcript during splicing): the resulting mRNA gives rise to internally-truncated but functional dystrophin protein. PIP6a PPMO molecules targeted to the splice donor site of exon 23 have been used to induce high levels of dystrophin in the *mdx* mouse: as shown previously [28], 10 doses of 12.5mg.kg$^{-1}$ PPMO at 2-weekly intervals results in widespread, uniform and high-level dystrophin protein expression (average ~50% of WT levels, with individual animals ranging from 30–80%) while a lower dose (6mg.kg$^{-1}$) results in substantially lower expression (average ~15% of WT, with similar variability–*D. Wells*, *manuscript in preparation*). Taking a single quadriceps muscle from each of these studies (high and low dose), we visualized PPMO-mediated induction of dp427 mRNA using RNAscope probes. As shown in Fig 9, the muscle treated with 6mg.kg$^{-1}$ PPMO (where skipping levels are low) appeared essentially indistinguishable from untreated *mdx* muscle, while muscle treated at 12.5mg.kg$^{-1}$ (where skipping levels are high) exhibited robust numbers of sarcoplasmic 5' and 3' foci. Particle analysis suggested these numbers were comparable to WT, and moreover showed 5'-3' co-localization not seen with lower dose treatment (S7A and S7B Fig). In this high dose treated muscle, the fraction of nuclei with large 5' foci was also restored to WT levels (25–30%) and nuclear 5' signal was similarly more intense, yet numbers of nuclei per field remained elevated (comparable with untreated *mdx*, see S7C Fig), with the corollary that all nuclear counts (5' and 3') were higher than those of WT muscle (i.e. implying supra-phyisological levels of nascent transcription within this muscle).

## Discussion

Resolution of mRNA molecules by *in-situ* hybridization has historically faced a number of challenges: abundant mRNAs can be detected relatively easily (indeed ISH using fluorescent probes (FISH) works well with such transcripts), but useful quantitative and spatial expression data is often lost in the consequent wash of strong signal. Conversely, low abundance transcripts (such as dystrophin) might well lend themselves to more nuanced analysis, but cannot usually be detected with significant confidence to allow such analyses. Most high-sensitivity ISH methods are either radiometric or colorimetric, and thus cannot easily be multiplexed. The RNAscope *in-situ* fluorescence amplification approach offers high sensitivity and multiplexing [24]: though typically used to resolve multiple mRNA species within a tissue, we show this ISH method can visualize multiple regions within a single transcript. The enormous genomic dystrophin (*DMD*) locus (~2.3Mbp, with 16-hour transcription time) further allows temporal metrics to be assessed, as nascent transcripts label with 5' probe only. Our work demonstrates a multiplex method that is particularly suited to long mRNAs transcribed from large genomic loci, allowing the identification of transcriptional subtleties that would otherwise remain intractable. Given that the dystrophin gene has key disease-relevance with many therapeutic approaches directed at the transcript level, our approach is particularly suited to its study.

### Subcellular localization of mature dp427 mRNA

We show that individual mature dp427 mRNAs can be detected within the muscle sarcoplasm, binding 5' and 3' probes in close proximity as expected. These mature transcripts are essentially absent in dystrophic muscle, yet comparatively high numbers (binding both probes) are

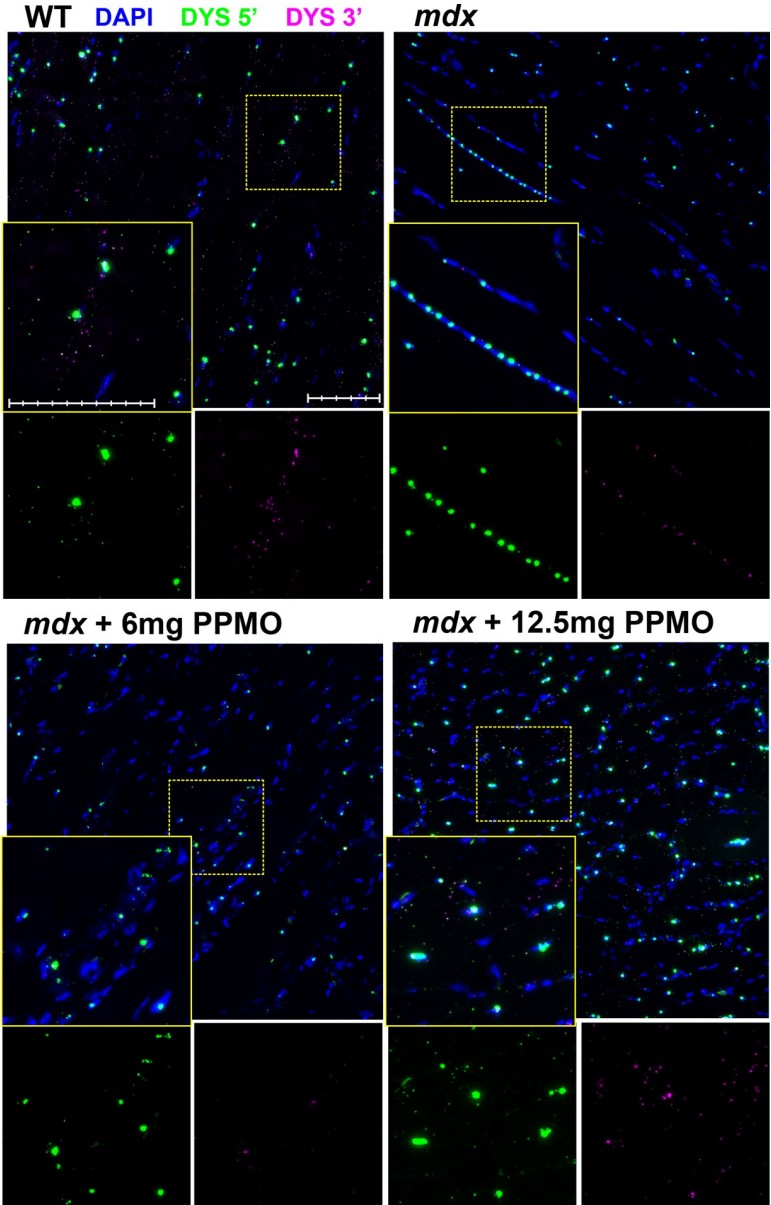

**Fig 9. RNAscope detection of dp427 restoration following PPMO exon skipping.** RNAscope probes to dp427 5'
(Cy3: green) and 3' (Cy5: magenta) in WT and *mdx* quadriceps muscle, or *mdx* muscle treated with PIP6a PPMO at
6mg.kg$^{-1}$ or 12.5mg.kg$^{-1}$ as indicated (note clear expression of nascent dp427 within centrally-located nuclei, *mdx*
inset). High doses of PPMO restore sarcoplasmic probe labelling. Scale bars: 100μm (main panel subdivisions: 20μm;
magnified panel subdivisions: 10μm).

found in rare patches that align with dystrophin immunoreactive revertant fibres. Previous *in-situ*
studies of dp427 expression suggested that this transcript preferentially localizes to the sarco-
lemma [17]. Given the sarcolemmal location of the dystrophin protein, this seems biologically
plausible. A single mRNA is relatively compact and can give rise to multiple protein molecules:
targeting dp427 mRNA to the sarcolemma before translation would be more efficient than trans-
porting protein post-translationally. Our data supports this: while in both longitudinal and trans-
verse orientations mature dp427 molecules are clearly found throughout the myofiber interior,
transcript concentration does appear to be highest immediately beneath the sarcolemma.

## Behaviour and quantification of mature dp427 mRNA

Under physiological conditions, most messenger RNA is expected to exhibit extensive secondary structure, and to be found as mRNP complexes with RNA binding proteins [43]. This is especially true for mRNA molecules with large 5' and 3' UTRs (such as dp427): the 5' and 3' termini of such mRNAs are reported to be very closely apposed in most cases (potentially in the order of nanometre separations [44]). Efficient probe hybridization however necessitates unfolding of mRNP complexes and mRNA secondary structure, potentially disrupting this co-localization. In healthy muscle labelled with our dystrophin probes, ~40% of sarcoplasmic 5' signals were nevertheless found within a micron of a 3' signal (and vice-versa), strongly suggesting that many transcripts remain close to their native configuration (and that the physiological state of a mature dp427 mRNA is indeed a condensed complex with 5' and 3' ends in close proximity). Remaining pair distances were also shorter than would be expected by chance, raising the possibility that some might represent labelling of unfolded transcripts. Length per base of single-stranded RNA remains contentious [45], and may vary with salt concentration and pH [46], thus we cannot empirically state the maximum permissible separation of probes within a single dp427 mRNA. Using reported values for ssDNA (~0.7nm per base [47]) would place the length of a mature full-length dystrophin transcript at ~10μm, with 5' and 3' probe binding sites separated by ~6μm (though ssRNA also adopts flexible, worm-like chain behaviour: mean 5'-3' distance would be substantially shorter [48]). The size and mobility of the amplifier 'trees' generated by RNAscope ISH is not reported, and deposits of fluorophore (activated tyramides react with tyrosine residues in adjacent proteins) might also extend beyond these limits, but restricting probe pairings to distances <6μm would account for ~70% of the observed foci in WT muscle (~80% with a more generous 8μm). Some foci will inevitably lie sufficiently out of focus as to be missed by particle analysis, and some dp427 transcripts will likely label with one probe but not the other (due to inhibitory secondary structure, masking by protein, or even cleavage of the molecule during sectioning): our mature mRNA counts might thus be an underestimate of true numbers, though our corroborating qPCR data suggests any underestimate is modest. Interestingly, counts of 3' foci were typically higher than 5', by a small but consistent amount (50–100). The consistent size and intensity of 3' foci allows this signal to be collected with greater ease than 5' (where overexposure of the strong nuclear foci must be balanced against underexposure of the weaker sarcoplasmic foci, and 5' foci in mature dp427 molecules close to nuclei may be lost amidst strong nascent 5' staining), however 3' foci without adjacent 5' might also represent labelling of dp71 [26]. While not reported to be expressed in muscle fibres specifically, this isoform is expressed in blood vessel endothelia [49] and proliferating myoblasts [50], and is detected at low levels within whole muscle tissue homogenates, particularly those from young dystrophic muscle undergoing acute degeneration/regeneration [51]. qPCR confirms dp71 is modestly expressed in both healthy and dystrophic quadriceps muscle (in numbers that might account for ~50–70 additional 3' foci per 20x image: see S8 Fig) but ISH also places this modest expression in context: dp71 is robustly expressed, but apparently restricted to a small number of specific cells, potentially only visible in dystrophic muscle where levels of dp427 are lower (see Fig 4B, lower panels). Endothelia are a plausible candidate, but combining dp427 multiplex ISH with cell-specific markers (such as PECAM) will be necessary to confirm this.

## Behaviour and quantification of nascent dp427 mRNA

The strong 5' probe signal within dp427-expressing nuclei is consistent with concerted expression (Smith *et al* reported similar findings in the nuclei of cultured myotubes using conventional fluorescent probes [25]: while unable to resolve single transcripts, they reliably resolved

the prominent nuclear dp427 transcriptional locus). This strong, unambiguous 5' nuclear labelling permits estimation of myonuclear number, a value otherwise difficult to determine. Muscle is host to multiple cell types, from blood-vessel endothelia and pericytes/mesoangioblasts to fibroblasts, adipocytes, and the dedicated muscle stem cell population of satellite cells [19, 20], but only myonuclei should robustly express full-length dystrophin and thus label strongly with 5' probe. Our ISH data indicates 20–30% of the nuclei within a typical non-tendinous region of healthy quadriceps muscle express dp427, while in dystrophic muscle this fraction is only 10–15% (despite similar absolute numbers of large 5' foci: see Table 1), commensurate with the higher fraction of non-myonuclear cell types within the dystrophic muscle environment. We also calculate that healthy dystrophin-expressing nuclei are typically host to 20–40 immature transcripts: if this represents steady-state occupancy, it implies active myonuclei produce one dp427 transcript every 20–30 minutes (though ~5 hours are required to transcribe the full 5' binding sequence, thus true nascent counts may be ~30% higher). Both ISH and qPCR data also suggest that mature dp427 mRNAs are in the clear minority: 60–80% of dp427 transcripts within healthy skeletal muscle are nascent, presumably arrayed along the dystrophin genomic locus (Fig 10A). The surprising corollary is that mature dp427 transcripts, despite taking 16 hours to produce, have half-lives of only ~2–3 hours (mean lifetime of 3–4 hours). Similar results were first reported in the late 90s by Tennyson and colleagues [16]

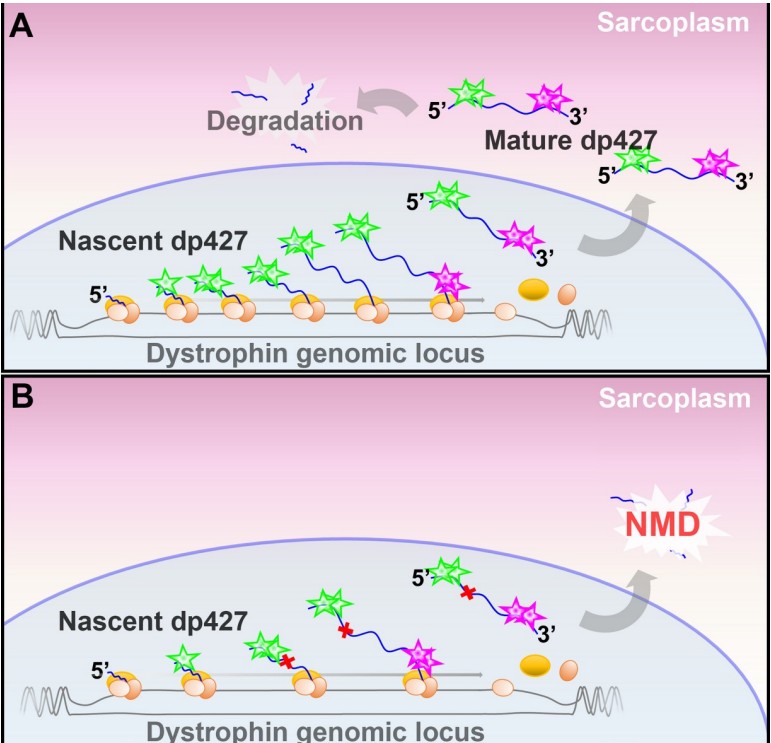

**Fig 10. Dystrophin transcriptional dynamics revealed by single-transcript multiplex ISH.** In WT muscle myonuclei (A), multiple transcriptional initiation events occur during the 16 hours needed for a single transcript to complete, resulting in a steady 'train' of RNA polymerases and nascent transcripts, and leading to high nuclear concentrations of dp427 5' sequence. Dp427 3' sequence is completed only ~15 minutes before transcript completion and export, thus is present at much lower levels within the nucleus. Mature dp427 mRNAs, validated by the pioneer round of translation and carrying both 5' and 3' sequence, are found distributed throughout the sarcoplasm but are then degraded. In *mdx* muscle myonuclei (B), transcriptional initiation events are less frequent, leading to fewer nascent 5' sequences within nuclei, and the presence of a PTC (red crosses) within the dp427 transcript leads to prompt post-export mRNA degradation via the NMD pathway.

using radiolabelled RT-PCR on samples of human skeletal muscle, and the values they obtained were remarkably close to those shown here: mean 5' and 3' totals were ~2500 and ~780 per ng RNA respectively (interestingly, with sample-to-sample variation similar to that found in our study), with concomitant derived half-life of ~3.5 hours. This early finding has garnered little mention since -even the authors themselves treated this result with scepticism- but our work here (showing similar values in a different species) strongly supports this remarkable conclusion. Tennyson *et al* reported a markedly longer half-life of ~16 hours in cultured human foetal myotubes: this discrepancy was attributed to either differential turnover rates between cultured myotubes and mature muscle, a higher rate of premature termination in mature muscle, or an indication that mature skeletal muscle was not at transcriptional steady-state (samples were collected from human cadavers). Our data, obtained from adult mouse muscle flash-frozen 2–5 minutes post-mortem, would argue against this latter hypothesis. Relatively high rates of premature termination are certainly plausible: at 2.3 million bases, the dystrophin gene likely tests the limits of RNA polymerase processivity [52], and some evidence exists for premature termination rates above predicted values [21], but such limits are a function of length, not maturity. It is thus unclear why rates would differ markedly between myogenic cultures and mature muscle. Dp427 may simply experience a higher rate of turnover in mature muscle: the demand for dystrophin is likely to be higher in nascent myotubes where the contractile environment is being established *de novo* (whereas turnover of an existing dystrophin protein pool may be all that is required in mature tissue). Production of dystrophin transcripts could be considered innately wasteful (synthesis of a 2.3Mb pre-mRNA for a final spliced transcript of only 14kb), but it nevertheless seems counterintuitive that mature muscle myonuclei dedicate 16 hours to producing a molecule with a mean subsequent lifespan of only 4. Dp427 transcripts comprise only a tiny fraction of myonuclear mRNAs (several thousand-fold lower than myosin heavy chain mRNA [16]) thus such waste is likely of little energetic consequence, and post-transcriptional control via mRNA stability in fact offers markedly greater utility than control of transcription itself: changes in expression based on transcriptional initiation would necessarily incur at least a 16-hour delay (and another 16-hour delay to return to basal levels) and this presumes the dystrophin locus is not already at near-maximal transcriptional occupancy. In contrast, changes in mRNA stability can be achieved rapidly via phosphorylation [53], and simply halting degradation of mature dp427 transcripts (leaving transcription unchanged) would double mRNA levels in less than half that time: constitutive overproduction paired with tuneable degradation is wasteful, but responsive. This mechanism might even be common among large genes: the human genome is host to a number of genes larger than 1Mb (>7 hour transcription times), many of which may need to alter expression over shorter timescales. Dynamic expression of dp427 would readily explain the high sample-to-sample variation reported here and previously [16], and control via mRNA stability would allow expression to fluctuate over a circadian timescale (as has been suggested [54]). It would be of great interest to examine the behaviour of our RNAscope probes in damaged (Cardiotoxin or BaCl$_2$-treated), regenerating healthy muscle: if stability of transcripts is higher in nascent myotubes, one might expect newly regenerated fibres to exhibit markedly higher levels of mature dp427 mRNA (something suggested by early ISH work [18]).

Tennyson *et al* derived estimates of 5–10 nascent dystrophin mRNAs per muscle nucleus [16]. Our data refines this estimate: not all nuclei within a muscle express dp427 (see above), but those that do are host to 20–40 nascent transcripts at a time. This argues in favour of a concerted commitment to dystrophin expression and provides further support for a maximized production hypothesis. Rare nuclei nevertheless appear to have lower (2–10) numbers of transcripts, suggesting that dystrophin-expressing nuclei might not all be at steady-state expression levels at any given moment: we cannot rule out a sustained but 'burst-like'

stochastic expression programme as is common to many eukaryotic genes [55] (and indeed transcription in muscle has been reported to exhibit 'pulsed' behaviour [56]). Transcriptional bursting could result in 'quiescent' myonuclei that do not label with 5' probe (leading to underestimation of myonuclear fraction). Expression of dp427 represents an implicit 16-hour commitment, however, and counts of dp427-expressing nuclei were moreover remarkably consistent between animals (Table 1 and Fig 5D), suggesting any underestimation is likely to be modest.

Dystrophic muscle reveals further nuances: while numbers of mature transcripts are reduced to almost zero as expected, the intensity of 5' probe labelling within myonuclei is also reduced. This reduction in total message is confirmed via qPCR. This latter observation could be dismissed as a reflection of the higher cell content and transcriptional diversity within dystrophic muscle (thus fewer total myonuclei), but our ISH data shows that this reduction in expression extends to the myonuclei themselves. Transcript degradation via NMD occurs only after transcription is complete [41], thus reduction in nuclear signal is best explained by a lower rate of transcriptional initiation (10–12 nascent transcripts per nucleus, thus one transcript initiated/completed every 60–90 minutes). It has been proposed that as a terminal marker of myogenesis, expression of dystrophin itself drives the myogenic programme to completion: low levels of dystrophin expressed earlier in differentiation help drive greater expression of dystrophin, further promoting terminal differentiation and producing a switch-like feed-forward loop [57, 58]. Dystrophic muscle, lacking this driver of terminal myofiber commitment, would thus be held in a permanent state of near-maturation, with concomitantly lower levels of dystrophin transcription and reduced 5' nuclear staining intensity (Fig 10B).

Our dp427 probes also detected transcripts within revertant fibres: these sporadic dystrophin-positive fibres are a long-recognized phenomenon [42, 59, 60], and are variably explained. Prevalence of revertants is influenced by the specific mutation carried (the PTC in exon 23 of the *mdx* mouse readily lends itself to revertant fibre formation, while the PTC in exon 53 of the *mdx*[4cv] mouse does not [61]), and revertant fibre number also tends to increase with age, clustering in a fashion that implies a clonal origin [62, 63] (skipped transcripts are found even in healthy muscle [64]). Current hypotheses tend to favour a coordinated, focally heritable alteration in splicing patterns rather than mutations at the genomic level [62]: our approach cannot distinguish these two proposals (and strong 5' labelling of all nascent mRNAs makes assignation of revertant expression to a specific nucleus or nuclei essentially impossible), but our data suggest that revertant fibres exhibit high (near-WT) levels of mature dp427 within relatively restricted domains, as might be expected if transcripts are short-lived. We also show ISH can detect corrected transcripts following therapeutic exon skipping, revealing spatial metrics that complement conventional assessments. While we examined only two muscles (one from a high dose treatment group, one from low dose), the muscle treated with a low dose of skipping oligonucleotide giving only 2–10% skipping efficiency (*D. Wells, manuscript in preparation*) was indistinguishable from untreated *mdx*, while the high dose treated muscle (which produces skipping efficiencies of 30–60% [28], and uniform sarcolemmal dystrophin protein) showed sarcoplasmic dp427 mRNA counts comparable to those of WT muscle. This data is only proof of principle (and the expense of this technique likely limits widespread diagnostic utility), but it illustrates potential limitations in measuring skipping percentages via qPCR alone, especially for an exon nearer the 5' end of the dp427 transcript (such as exon 23): values expressed as a ratio do not give absolute numbers, and while skipped transcripts will give rise to mature dp427 mRNAs (unlike unskipped, which will be degraded), even in healthy muscle, mature mRNA represents only ~20% of total dp427; the majority of the dp427 measured via qPCR (skipped or unskipped) will be nascent rather than mature transcripts. ISH conversely allows direct visualisation and quantification of mature transcripts (and estimation of total transcript numbers), but does not permit assessment of skipping percentages. This

preliminary data further suggests multiplex ISH might reveal changes in transcriptional dynamics following treatment: total nuclei per field in the 12.5mg.kg$^{-1}$ PPMO (high dose) treated muscle were higher than WT (comparable to untreated *mdx*) but the fraction of nuclei expressing dp427 was comparable to healthy muscle (i.e. total numbers of dystrophin-expressing nuclei were higher than WT); even without 100% skipping efficiency, treated muscles might achieve physiological levels of mature dp427 by producing supra-physiological levels of nascent transcripts. Questions remain, however: skipped transcripts are typically detectable within a week of systemic treatment, but a study following long-term response to exon skipping treatment in the *mdx* mouse [65] reported a 2–3 week lag between first detection of skipped transcripts and restoration of dystrophin protein, and further showed that both skipped dp427 transcripts and particularly dystrophin protein itself persisted for some weeks following the cessation of treatment. Levels of skipped transcripts and dystrophin protein in this study were low (~10% and 1–3% of WT respectively) but these data nevertheless suggest that the mean lifespan of dystrophin protein is of the order of days to weeks, and moreover that protein turnover is comparatively slow. It is hard to reconcile this with the mRNA dynamics revealed by our studies here (and those reported historically [16]): despite a lengthy transcription time, mRNA turnover of dp427 appears to be remarkably rapid. Tissue persistence of antisense oligonucleotides was found to decline in line with falling skipping percentages [65], compatible with ongoing skipping of short-lived dp427 mRNAs, but the high stability of the dystrophin protein and the fact that this protein represents only a relatively minor component of the muscle protein milieu [14] seems incompatible with such dynamic transcript behaviour. The implication is that in healthy muscle, dp427 mRNA molecules may be actively degraded despite having contributed essentially no dystrophin protein beyond the first post-export pioneering ribosomal proof-read: at the mRNA level, supply might substantially exceed demand. If correct, this surprising finding might be of considerable value to therapeutic approaches for DMD such as gene editing: if under healthy conditions many more transcripts are produced than are needed, in dystrophic conditions beneficial levels of dystrophin protein might be restored with only modest levels of genomic correction (and studies suggest this may be the case [66–70]).

## Concluding remarks

The work shown here validates a novel, spatiotemporal use of RNAscope methodology, demonstrating the value of multiplex targeting within a single transcript, particularly with mRNAs transcribed from long genomic loci such as the dystrophin gene. In this study we focus on mature mouse skeletal muscle, a tissue within which dystrophin expression is almost exclusively dp427. Single transcript multiplex ISH (5' and 3') reveals dystrophin transcriptional dynamics, showing that even in healthy muscle most dp427 is nascent, and that mature transcripts are remarkably short-lived (a conclusion supported by qPCR). Subcellular resolution of individual transcripts -and moreover discrete regions of those transcripts- has wider applications. Use of a third 'middle' probe alongside the 5' and 3' probe sets described here allows the full-length dp427 isoform to be distinguished from dp260/dp140, and in turn from dp116/dp71: we have used this ISH method to study expression of dystrophin isoforms during embryonic development [26], something historically precluded by shared sequence identity at both protein and mRNA level. A logical target for further investigations would be the brain, a complex tissue known to express a similarly complex combination of dystrophin isoforms: our ISH approach would allow these isoforms to be mapped to subcellular levels and might allow us to determine whether brain dp427 isoforms (dp427c, dp427p) are as short-lived as dp427m. Indeed, while the current therapeutic focus is on skeletal muscle, the brain is also known to be

affected by dystrophin mutations: if treatment of skeletal muscle continues along current encouraging lines, the brain might be the next major therapeutic target, and the *in situ* hybridization methodology described here offers a powerful tool to aid such investigations.

## Supporting information

**S1 Fig. Fresh-frozen muscle does not tolerate the detergent incubations used in the RNA-scope hybridization/washing stages without additional fixation and baking.** Serial slides were treated according to the protocol for fresh-frozen tissue, with a single slide taken at each indicated step and stained via Haematoxylin and Eosin. Protease treatments are well-tolerated, but muscle is visibly paler after probe incubation (lower left) and all myofibrillar material is lost entirely after a single subsequent wash.
(TIF)

**S2 Fig. Positive and negative control probes.** Positive control probes (Polr2a, Ppib, UBC -upper panels): Both WT (A) and *mdx* (B) muscle show the expected pattern: rare foci are observed for probes to Polr2a (very low abundance transcript in muscle), Ppib (low abundance in muscle) produces more foci, and staining for UBC (high abundance) is widespread. All transcripts appear to be more plentiful immediately beneath the muscle sarcolemma. Dystrophic muscle also shows patches of prominent non-specific staining (particularly in FITC and Cy3 channels) which may correspond to peroxidase-rich macrophage/neutrophil infiltrates (arrowheads). Negative control probes (bacterial DapB): WT muscle (C) shows no staining in any of the negative control probe channels. Dystrophic muscle (D) shows no probe-specific foci, but as with positive control staining (above), this tissue exhibits patches of prominent non-specific staining (particularly in FITC and Cy3 channels) which may correspond to macrophage/neutrophil infiltrates (arrowheads). Scale bars: 100μm (main panel subdivisions: 20μm; magnified panel subdivisions: 10μm).
(TIF)

**S3 Fig. RNAscope labelling of dp427 5' and 3' with reversed fluorophores.** RNAscope probe labelling of 40-week old WT and mdx quadriceps muscle. Probe to dp427 5' (Cy5: magenta) resolves both small sarcoplasmic foci and large nuclear foci, while 3' probe (Cy3: green) shows small foci only, showing that this pattern is a property of the probes, not the fluorophores selected. Scale bars: 100μm (main panel subdivisions: 20μm; magnified panel subdivisions: 10μm).
(TIF)

**S4 Fig. Foci per image correlates with oxidative capacity and nuclear count.** (A) total 5' and 3' counts per image, per individual (9–30 images per animal). (B) Alignment of RNAscope images collected with oxidative capacity: DAPI channel of RNAscope-probed WT muscle with locations of images indicated (left) and matching approximate locations in an SDH-stained serial section (right). Strongly SDH-stained regions (1) have more foci, but also more nuclei than regions of less oxidative fibres (2). SDH intensity correlates with total foci count but does not reach significance (C), while total foci count and number of nuclei per field show greater correlation (D). Images collected from *mdx* sections have significantly higher numbers of nuclei per field (E). Correlations: Spearman's rho. Nuclear counts: Mann-Whitney U test using mean per image nuclear counts, per animal (N = 4 per genotype).
(TIF)

**S5 Fig. Numbers of small 5' and 3' foci per image show strong correlation in WT muscle, but not necessarily in dystrophic muscle.** (A) Representative plot of 5' counts vs 3' counts for 1 healthy and 1 dystrophic individual. (B) Correlations per individual for both small foci and

total foci (top rows: Pearson r values; Bottom rows: significance. * = P<0.05;** = P<0.005;*** = P<0.0005;**** = P<0.0001; N/S = not significant). (C) Distributions of nearest-neighbour distances greater than 1μm in images from WT (left) and *mdx* (right) muscle, compared with random distributions of the appropriate number of points (WT: 800, 1200; *mdx*: 200, 400) or from the same images after 3' channel rotation. Distances <4μm are over-represented in both WT and *mdx* muscle.
(TIF)

**S6 Fig. Large and small 5' foci can be quantitatively compared.** Left columns: Images of 5' foci only were collected at 100, 200, 400, 600 and 800ms exposure times and the highest exposure images were used to define regions of interest (large foci, small foci, background). Mean fluorescence intensity in each ROI was determined for each exposure time and plotted (right). Background and small foci fluorescence values increase linearly with exposure time, whereas large foci show attenuation above 200msec, showing signal saturation. All comparisons thus used 100 and 200msec values. Images shown is cropped for clarity: all analysis used full size images with 30–50 foci of each class defined per image, using 3–5 images per individual.
(TIF)

**S7 Fig. Additional exon-skipping analyses.** (A) Numbers and distributions of 5' and 3' foci in PPMO treated muscles. WT and untreated mdx values are from Fig 5. Treatment with 12.5mg.kg$^{-1}$ restores small sarcoplasmic dystrophin counts to near-WT levels but also increases total nuclear counts of all foci. (B) PIP6a PMO treatment at 6mg.kg$^{-1}$ does not restore probe co-localization whereas treatment at 12.5mg.kg$^{-1}$ does. (C) PIP6a PPMO-treated muscles still exhibit per-image nuclear counts comparable to untreated mdx muscle.
(TIF)

**S8 Fig. Expression of dp71 in skeletal muscle.** Absolute quantification of transcript numbers in WT and *mdx* quadriceps muscle using primers to exons 62–63 of dp427 (mature dp427) or to exons 1–2 of dystrophin isoform dp71 (dp71 unique first exon, and exon 63 of dp427). Dp71 is expressed in muscle but at low levels (50–70 transcripts per ng RNA) and is not altered in response to dystrophic pathology. Numbers of dp71 transcripts are however comparable with the greatly reduced numbers of mature dp427 transcripts in *mdx* muscle.
(TIF)

## Acknowledgments

This manuscript was approved by the RVC research office and assigned the following number: CSS_02001.

## Author Contributions

**Conceptualization:** John C. W. Hildyard, Richard J. Piercy.

**Data curation:** John C. W. Hildyard.

**Formal analysis:** John C. W. Hildyard.

**Funding acquisition:** Richard J. Piercy.

**Investigation:** John C. W. Hildyard, Faye Rawson.

**Methodology:** John C. W. Hildyard, Faye Rawson.

**Project administration:** Richard J. Piercy.

**Resources:** Dominic J. Wells.

**Software:** John C. W. Hildyard.

**Supervision:** Richard J. Piercy.

**Visualization:** John C. W. Hildyard.

**Writing – original draft:** John C. W. Hildyard.

**Writing – review & editing:** John C. W. Hildyard, Faye Rawson, Dominic J. Wells, Richard J. Piercy.

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
