## [Decision Letter · Decision Letter 0]

17 Jun 2020

PONE-D-20-09506

Multiplex *in situ* hybridization within a single transcript: RNAscope reveals dystrophin mRNA dynamics

PLOS ONE

Dear Dr. Hildyard,

Thank you for submitting your manuscript to PLOS ONE. After careful consideration, we feel that it has merit but does not fully meet PLOS ONE’s publication criteria as it currently stands. Therefore, we invite you to submit a revised version of the manuscript that addresses the points raised during the review process.

We look forward to receiving your revised manuscript.

Kind regards,

Atsushi Asakura, Ph.D

Academic Editor

PLOS ONE

Journal Requirements:

2. Your ethics statement must appear in the Methods section of your manuscript. If your ethics statement is written in any section besides the Methods, please move it to the Methods section and delete it from any other section. Please also ensure that your ethics statement is included in your manuscript, as the ethics section of your online submission will not be published alongside your manuscript.

Reviewers' comments:

Reviewer's Responses to Questions

**Comments to the Author**

1. Is the manuscript technically sound, and do the data support the conclusions?

Reviewer #1: Partly

Reviewer #2: Yes

2. Has the statistical analysis been performed appropriately and rigorously? 

Reviewer #1: Yes

Reviewer #2: No

3. Have the authors made all data underlying the findings in their manuscript fully available?

Reviewer #1: Yes

Reviewer #2: Yes

4. Is the manuscript presented in an intelligible fashion and written in standard English?

Reviewer #1: Yes

Reviewer #2: Yes

5. Review Comments to the Author

Reviewer #1: In this manuscript, the authors present a glimpse into the complex transcriptional dynamics of the dystrophin gene using RNAscope, a fluorescent in situ hybridization technique. Utilizing the significant size of the gene and its known (lengthy) transcription time, probes against distinct 5’ and 3’ regions within the gene allow for the visualization of nascent versus mature RNAs in a spatial context. Both healthy and dystrophic muscle is analyzed, as well as muscle from dystrophic mice that have undergone therapeutic intervention. The data, while interesting, are not necessary novel; RNAscope is a technique that was created for histological use, the transcriptional dynamics of dystrophin observed in the paper have been described already (albeit in systems with far lower resolution), and the distinction of nascent versus mature transcripts has been shown via live-imaging. The novelty of this manuscript therefore lies on the visualization of these dynamics in fixed muscle samples and the demonstration of this technique as a potential way to assess therapeutic efficacy. I feel this point could be further emphasized and clarified, and have suggested some experiments below.

Despite the lack of significant novelty, the data presented are clear and robust, and the authors have provided significant detail on the methodology, image quantification, and analysis – all of which are crucial for reproducibility and widespread adoption of the technique. With some further characterization, I feel the manuscript and the methods described within it will provide a useful tool for the muscular dystrophy field.

Specific comments:

1. As mentioned in the introduction of this manuscript and depicted in Figure 1, there are a number of truncated isoforms of dystrophin. Dp71, is present in the liver, lung, kidneys, and brain (Bar et al., 1990) – but not muscle. Histological RNAscope analysis on those tissues should therefore yield only signal from the 3’ probe and serve as a necessary control to demonstrate the specificity of the RNAscope probes.

2. Dp427 RNAs produced by mdx mice contain a PTC in exon 23, and therefore the region recognized by the 3’ probe shouldn’t be transcribed. Why, then, are there so many 3’ puncta in mdx muscle? If these foci are indeed labeling interstitial cells, as suggested in the discussion, the manuscript would benefit from adding data to distinguish between cell types within the tissue. This could be done through the multiplexing of immunohistochemistry or merely the addition of cell-specific RNAscope probes on other channels (PDGFRa, PECAM, etc.).

3. In the discussion (page 30), the authors state that “in healthy muscle, myonuclei comprise 20-30% of the nuclei within a typical non-tendinous region […], while in dystrophic muscle this figure is 10-15%.” These numbers are significantly lower than have been reported in the skeletal muscle field – how were these percentages calculated? Outlining of myofibers via IHC and subsequent quantification is needed to support this claim.

4. While the reported numbers of small foci remain consistent throughout the figures, the representative images are visually inconsistent. The mdx panels in Figure 9 show very few 3’ foci, all within myonuclei (as would be more consistent with early termination); Figure 4B, however, shows noticeably more 3’ foci both within and outside of myonuclei.

5. The PPMO-treated mouse experiments shown in Figure 9 were only performed with a n=1. While we understand that these experiments are likely expensive and time-consuming, at least a n=2 is necessary to make any conclusions on the efficacy of these treatments. An additional minor point: in Figure 9, “PMO” is written above the fourth image set instead of “PPMO”

6. Improved labeling is needed in figures to distinguish between images of healthy vs dystrophic muscle sections.

7. At 10 pages, the Discussion is overly long and would benefit from shortening.

Reviewer #2: The authors provide a thorough analysis of the dystrophin transcript using RNA scope in tissue sections from WT and mdx mice. The authors use probes recognizing the 5’ and 3’ end of the dystrophin mRNA to estimate the amount of full-length transcripts within WT and dystrophic myofibers. While some of the data generated confirms previously published observations, some new findings, including the increased amount of dystrophin transcript in PPMO-treated mdx animals, is interesting and novel. There are some points that could be better clarified to strengthen the study, as listed below.

1) It is reported that the dystrophin transcript localizes near the fiber membrane. While this is intuitive because of the location of myonuclei in fibers, did the authors observe any probe signal (5’ or 3’) in centrally located nuclei, which are very common in mdx mice?

2) Figure 7 revertant fiber: the image shows expression of dystrophin 3’ mRNA in nuclei that appear to be outside the membrane boundary of the revertant fiber. The authors should provide a better example of the 3’ probe signal being evident within revertants.

3) Figure 9. Please provide a consecutive section stained for dystrophin protein to corroborate the RNA data, showing that with high levels of PPMO, dystrophin protein is detected. The animal was treated for many weeks and protein expression should be detectable.

4) The PPMO data is gathered from a single mouse, single muscle. It is felt that this seems to be a limited observation overall that should be further substantiated. At the very least, multiple muscles should be analyzed from the mouse to confirm the findings seen in the quadriceps.

5) Similarly, in Figure 8 determinations were made by analyzing a total of 60-120 particles (page 23) per mouse, which seems rather low to draw robust conclusions.

6. PLOS authors have the option to publish the peer review history of their article (what does this mean?). If published, this will include your full peer review and any attached files.

Reviewer #1: No

Reviewer #2: No

---

## [Author Response · Author response to Decision Letter 0]

12 Aug 2020

Response to reviewers for manuscript

PONE-D-20-09506

Multiplex in situ hybridization within a single transcript: RNAscope reveals dystrophin mRNA dynamics

We thank the editor and the reviewers for their helpful comments and constructive suggestions. We have made several edits to the manuscript to address the concerns raised, and feel that the text is markedly improved as a consequence. Given the restrictions imposed by the ongoing pandemic (and the accompanying uncertainly), further supporting experiments will be extremely challenging to conduct over a practical timescale: accordingly, we have instead moderated our wording to emphasise the proof-of-principle nature of some of our observations. We hope this will satisfy the reviewers. Furthermore, within the original manuscript we also made reference to potential future work exploring the possibility of using three probes to dp427, allowing shorter dystrophin isoforms (d140, dp71) to be detected, and indeed reviewer #1 raised some excellent questions regarding this possibility. In the period between submission of this manuscript and reviewer responses, this work has been published (https://wellcomeopenresearch.org/articles/5-76/v2). We have therefore necessarily made several more substantial edits to our manuscript to reflect the progression of this work from ‘future proposal’ to ‘published findings’: fortuitously, these changes both address the question raised by reviewer #1 and shorten the discussion slightly (as requested). 

Specific responses to the editor:

We have edited our manuscript to match the suggested style requirements: edits include

• Changing capitalization and font size on headings and subheadings

• Changing author name format to “first name initials surname”

• Reformatting table 1

• Tab indenting all paragraphs and reformatting methods subheadings

• Reformatting of figure titles and legends 

2. Your ethics statement must appear in the Methods section of your manuscript. If your ethics statement is written in any section besides the Methods, please move it to the Methods section and delete it from any other section. Please also ensure that your ethics statement is included in your manuscript, as the ethics section of your online submission will not be published alongside your manuscript.

Our ethics statement did indeed appear in our manuscript, and was indeed within the methods section (within the sample preparation subsection). To remove ambiguity, we have edited the methods to assign a unique subheading specifically to our ethics statement, and have also updated the ethics statement within the submission system accordingly.

Specific responses to reviewers:

Reviewer #1: In this manuscript, the authors present a glimpse into the complex transcriptional dynamics of the dystrophin gene using RNAscope, a fluorescent in situ hybridization technique. Utilizing the significant size of the gene and its known (lengthy) transcription time, probes against distinct 5’ and 3’ regions within the gene allow for the visualization of nascent versus mature RNAs in a spatial context. Both healthy and dystrophic muscle is analyzed, as well as muscle from dystrophic mice that have undergone therapeutic intervention. The data, while interesting, are not necessary novel; RNAscope is a technique that was created for histological use, the transcriptional dynamics of dystrophin observed in the paper have been described already (albeit in systems with far lower resolution), and the distinction of nascent versus mature transcripts has been shown via live-imaging. The novelty of this manuscript therefore lies on the visualization of these dynamics in fixed muscle samples and the demonstration of this technique as a potential way to assess therapeutic efficacy. I feel this point could be further emphasized and clarified, and have suggested some experiments below.

Despite the lack of significant novelty, the data presented are clear and robust, and the authors have provided significant detail on the methodology, image quantification, and analysis – all of which are crucial for reproducibility and widespread adoption of the technique. With some further characterization, I feel the manuscript and the methods described within it will provide a useful tool for the muscular dystrophy field.

We thank the reviewer for this summary. We agree this technique has potential therapeutic value: ability to discern mature (corrected) dp427 transcripts from the more numerous nascent transcripts (corrected and uncorrected) enhances our understanding of exon-skipping based therapies. The ability to further resolve transcript distributions within muscle adds a spatial element otherwise lacking from current investigations. 

This ISH approach is however expensive, and we expect initial assessments of therapeutic efficacy will continue to employ more conventional approaches (such as PCR). We have added words to this effect to the discussion.

We instead place greater emphasis on our broader findings, as we feel these reveal fascinating elements to dystrophin transcript dynamics that would be challenging to identify by other means (and which may be of interest to a wider audience). As we address in our discussion, the values we obtain for mRNA half-life (~3-4hrs) are counter-intuitive, given the ~16-hour transcription time. Mature dp427 transcripts are short-lived: most dystrophin mRNA within skeletal muscle appears to be nascent. Such discrepancies suggest that control of dystrophin expression is predominantly post-transcriptional, and as we further note this “maximise production, degrade according to demand” approach might be a more general strategy for long genes where rapid transcriptional responses are not viable. This finding therefore has ramifications that are pertinent for transcriptional modifier approaches in therapeutics.

The reviewer claims these findings are not novel: we note that novelty per se is not a requirement for publication in PLOSone, but we would nevertheless be extremely grateful if the reviewer could provide specific citations to support this statement. While elegant dp427 mRNA-labelling experiments have certainly been performed (particularly in zebrafish), we are unaware of published approaches similar to ours (especially in mammalian systems). With respect to half-lives specifically, we acknowledge that Tennyson et al obtained very similar values in human muscle in 1996, and indeed we cite these studies extensively. The possibility that dp427 mRNA might experience rapid turnover has nevertheless remained controversial. Given the apparent high stability of the dp427 protein (for which a considerable body of literature exists), the general perception appears to be that transcript stability is similarly high (as we note, even Tennyson et al regarded their values with some scepticism). Our work supports these historical findings, using a different approach, in a different mammalian system. We discuss the implications of such transcript turnover in health and disease, and the possible impact on current therapeutic intervention strategies. We therefore feel the data are novel.

Specific comments:

1. As mentioned in the introduction of this manuscript and depicted in Figure 1, there are a number of truncated isoforms of dystrophin. Dp71, is present in the liver, lung, kidneys, and brain (Bar et al., 1990) – but not muscle. Histological RNAscope analysis on those tissues should therefore yield only signal from the 3’ probe and serve as a necessary control to demonstrate the specificity of the RNAscope probes.

An excellent point, thank you. We have recently published a more comprehensive study addressing this pertinent observation, demonstrating expression of multiple isoforms in embryonic tissues via a triplex ISH approach (as proposed in our original discussion). 

https://wellcomeopenresearch.org/articles/5-76/v2

These studies strongly support our findings here: signal consistent with dp71 is found in locations known to be rich in this isoform, and the spatial resolution of this technique is such that isoforms can be distinguished even when expressed in closely-associated tissue types (the epithelium of the developing lung is rich in 3’ probe signal only, while the underlying bronchiole smooth muscle gives labelling consistent with dp427).

At the time this manuscript was submitted, the further work described above was unpublished. Given the reviewer’s suggestion, we have edited the text throughout to reference this additional study (and to reflect on the findings therein) where appropriate.

2. Dp427 RNAs produced by mdx mice contain a PTC in exon 23, and therefore the region recognized by the 3’ probe shouldn’t be transcribed. Why, then, are there so many 3’ puncta in mdx muscle? If these foci are indeed labeling interstitial cells, as suggested in the discussion, the manuscript would benefit from adding data to distinguish between cell types within the tissue. This could be done through the multiplexing of immunohistochemistry or merely the addition of cell-specific RNAscope probes on other channels (PDGFRa, PECAM, etc.).

With all respect, the claim that there are many 3’ foci in dystrophic muscle is incorrect: numbers of 3’ foci are markedly lower (3-4 fold) in this tissue. Moreover, as we show, this reduction is exclusively in sarcoplasmic foci. 

Premature termination codons do not prevent transcription (or indeed splicing) of a gene: the gene is transcribed as normal but then (following pioneer translation upon nuclear export) subsequently targeted for nonsense mediated decay (NMD). The region targeted by the 3’ probe emerges late in transcription (~1hr before completion) thus we would not expect to observe strong, large nuclear foci (as seen with 5’ probe), but we would still expect to see some nuclear 3’ labelling (transcripts nearing completion but not yet exported). As our data shows, small nuclear 3’ foci are indeed present within mdx muscle at numbers comparable to healthy muscle. The finding is entirely consistent with what is expected for a transcript that contains a PTC.

We accept that the numbers of sarcoplasmic 3’ foci are not zero in dystrophic muscle, and we address this observation within our manuscript: some of these will be mature dp427 transcripts within revertant fibres, and these are typically accompanied by a similar number of small sarcoplasmic 5’ foci as would be expected (we refer the reviewer to figure 7). Other 3’ foci (not found in close apposition with 5’ foci) are best explained as dp71 transcripts, a hypothesis supported by our observations of known dp71-expressing lineages in embryonic development (as described above), and further supported here by the observation that in muscle these foci are often found clustered, associated with single nuclei (see figure 4B), rather than scattered randomly in a manner more consistent with background noise. We know this short isoform is expressed in skeletal muscle at low levels (we measure this via qPCR: see supplementary figure 8), and in muscle of the ages studied, numbers of dp71 transcripts appear comparable between healthy and dystrophic muscle. It is consequently likely that a small number of 3’ foci found in healthy muscle also correspond to dp71 transcripts, but detecting these unambiguously amidst high numbers of dp427 transcripts is likely impossible. Only in mdx muscle, where mature dp427 transcripts are absent (revertants excepted), can these rare dp71 expressing cells be identified. We have edited the text to make this clear to the reader.

“qPCR confirms dp71 is modestly expressed in both healthy and dystrophic quadriceps muscle (in numbers that might account for ~50-70 additional 3’ foci per 20x image: see Supplementary Fig S8) but ISH also places this modest expression in context: dp71 is robustly expressed, but apparently restricted to a small number of specific cells, potentially only visible in dystrophic muscle where levels of dp427 are lower.”

The suggestion to add IHC for specific cell markers is excellent, and we wholly agree such investigations would be of interest. Dual ISH/IHC is not a trivial exercise, however: RNAscope ISH requires sample fixation (for fresh frozen muscle, this is an extended fixation, followed by a baking step), precluding the use of many native epitope antibodies typically used in fresh frozen muscle (antibodies to dystrophin are particularly fussy in this respect, in our experience). Furthermore, use of DAPI, cy3 and cy5 (for nuclei, 5’ and 3’ probes respectively) leaves us with only a single remaining fluorophore channel available, allowing us (at most) to investigate a single antibody/marker at a time, at considerable cost in time and consumables, with no guarantee of success. Use of additional RNAscope probe sets would be more tractable, albeit no less costly: were resources infinite this would indeed be an elegant series of experiments, but we are necessarily limited in this respect. Determination of muscle cell-types expressing dp71 could form the basis of future investigations, but we feel the specific identity of these cells is not germane to the conclusions of this current manuscript: we instead elect to suggest possible candidates rather than make unwarranted assumptions. Given the reviewer’s suggestions, we have edited our discussion to address this question directly. 

“Endothelia are a plausible candidate, but combining dp427 multiplex ISH with cell-specific markers (such as PECAM) will be necessary to confirm this.”

3. In the discussion (page 30), the authors state that “in healthy muscle, myonuclei comprise 20-30% of the nuclei within a typical non-tendinous region […], while in dystrophic muscle this figure is 10-15%.” These numbers are significantly lower than have been reported in the skeletal muscle field – how were these percentages calculated? Outlining of myofibers via IHC and subsequent quantification is needed to support this claim.

The calculation is simple: as we state in our discussion (on page 30), only myonuclei should express full-length (dp427) dystrophin. Since our method readily identifies these nuclei by their prominent 5’ labelling, and total numbers of nuclei per imaging field can similarly be determined by automated counting, the fraction of myonuclei is simply the former divided by the latter. We accept that “fraction of nuclei expressing dp427” is a more accurate descriptor, and have edited the manuscript accordingly. 

As we later note (page 32) we cannot rule out transcriptional bursting: only myonuclei should express full-length dystrophin, but this does not imply they do so constitutively. Our estimate may thus be lower than true values, but we do not believe it to be markedly so: once committed to expressing dp427, a nucleus remains committed for at least 16 hours, even if only producing a single transcript. As we show, dystrophin-expressing nuclei are typically host to multiple nascent transcripts (20 or more), implying a yet longer commitment. Transcriptional bursting may occur, but the lengthy dp427 transcription time precludes rapid switching between expressing/non-expressing states. We have edited the text to clarify our reasoning here. 

The reviewer implies that published values are significantly higher: again a citation to support these comments would be much appreciated. A fairly extensive search of the literature identifies a number of studies, but none seems to address this specific question. Many studies investigate either isolated myofibres (from which nuclei will almost exclusively be myonuclei), or isolated nuclei (which again will predominantly be myonuclei, as the disruption methods used are typically too mild to liberate nuclei from mononuclear cells). We surmise that this question is simply rather challenging to answer, and our approach here offers one approach by which a value might be obtained. We note that an elegant recent study by Hastings et al (https://skeletalmusclejournal.biomedcentral.com/track/pdf/10.1186/s13395-020-00233-6) suggests another means by which this question might be answered, but beyond acknowledging that muscle is host to large numbers of non-muscle cell types, the authors do not report specific percentages.

4. While the reported numbers of small foci remain consistent throughout the figures, the representative images are visually inconsistent. The mdx panels in Figure 9 show very few 3’ foci, all within myonuclei (as would be more consistent with early termination); Figure 4B, however, shows noticeably more 3’ foci both within and outside of myonuclei.

We thank the reviewer for giving our figures such detailed examination. The imaging field shown in figure 4B was specifically selected to show the rarely-observed 3’ labelling nuclei (consistent with dp71 expression). We freely admit within our manuscript that dystrophic muscle exhibits greater image-to-image variation in counts of probe foci. Cells expressing dp71 (3’ probe only) are found in some images, not in others. Regions consistent with revertant fibres (small sarcoplasmic foci of both probes) are also observed rarely, as would be expected. The numbers shown in figures represent the mean of multiple images collected from each individual (see methods), not per-image values.

5. The PPMO-treated mouse experiments shown in Figure 9 were only performed with a n=1. While we understand that these experiments are likely expensive and time-consuming, at least a n=2 is necessary to make any conclusions on the efficacy of these treatments. 

Our intent was to demonstrate detection of exon-skipped dp427 mRNA in treated dystrophic muscle: while we suggest our approach could indeed be used to assess therapeutic efficacy, we do not attempt this assessment within the manuscript. As the reviewer rightly notes, our sample size is insufficient for any such assessment, and we acknowledge this. The muscles used were taken from two studies investigating long-term PPMO treatment: one of these studies is published, the other is currently in preparation for submission. As such, the efficacy of these treatments is already known (measured via qPCR for skipping efficiency at the mRNA level, and via western/IHC at the more crucial protein level): 12.5mg.kg-1 is highly effective, while 6mg.kg-1 is not (N=6-7 per treatment group). Our data (with limited sample size) agrees with these published findings. The spatial nascent/mature distinction offered by our method complements, rather than replaces, these assessments: qPCR for skipped/unskipped transcripts does not distinguish between nascent and mature mRNAs, while 5’ and 3’ probes do not distinguish between skipped and unskipped transcripts (exon 23 lies outside both probe target regions). We have edited the discussion (see page 34) to make this more obvious to the reader.

Further experiments would be highly challenging at this point (and would add little to our conclusions), thus we have instead moderated our wording to refer to ‘detection of rescued transcripts’ rather than ‘assessment of therapeutic efficacy’, and have moved the bar charts from figure 9 to the supplementary material to illustrate that our primary assessment is chiefly qualitative. 

An additional minor point: in Figure 9, “PMO” is written above the fourth image set instead of “PPMO”

Thank you for spotting this. We have corrected this figure.

6. Improved labeling is needed in figures to distinguish between images of healthy vs dystrophic muscle sections.

We apologise. We have edited the figures to make it clear in each instance whether the muscle is WT or mdx, and similarly edited the figure legends.

7. At 10 pages, the Discussion is overly long and would benefit from shortening.

We have made efforts to be concise but the conclusions of the paper and the technical and novel nature of the methodology means that detailed descriptions are required to ensure a reader interprets the paper appropriately. We have made edits where possible, but are happy to consider further trimming specific areas if the reviewer or editor feel this is necessary and can advise. Notably, some future work described in the conclusion is now published: accordingly several passages (including the conclusion) have been substantially reduced. 

Reviewer #2: The authors provide a thorough analysis of the dystrophin transcript using RNA scope in tissue sections from WT and mdx mice. The authors use probes recognizing the 5’ and 3’ end of the dystrophin mRNA to estimate the amount of full-length transcripts within WT and dystrophic myofibers. While some of the data generated confirms previously published observations, some new findings, including the increased amount of dystrophin transcript in PPMO-treated mdx animals, is interesting and novel. There are some points that could be better clarified to strengthen the study, as listed below.

We thank the reviewer for their concise summary of our work, and their detailed assessment. We address the reviewer’s specific questions accordingly, below. 

1) It is reported that the dystrophin transcript localizes near the fiber membrane. While this is intuitive because of the location of myonuclei in fibers, did the authors observe any probe signal (5’ or 3’) in centrally located nuclei, which are very common in mdx mice?

An excellent question: thank you. As the reviewer notes, centrally nucleated fibres are common to dystrophic mice: a unique feature of this animal model is that such central nuclei persist, rather than gradually adopting a peripheral location. In mdx quadriceps of this age, very few myofibres will not be centrally nucleated, but dp427 is expressed nevertheless (even if mature transcripts are degraded via NMD). The answer to this excellent question is thus “almost certainly”. Note that unlike the healthy muscle sections, where myonuclei are scattered along the (likely) myofiber periphery, in mdx muscle long chains of closely-aligned nuclei are present: a location consistent with the myofiber centre (figure 9 provides an excellent example of this). As shown, these nuclei certainly contain nascent dystrophin transcripts (strong nuclear 5’ labelling).

We have edited the text to reflect this pertinent observation.

2) Figure 7 revertant fiber: the image shows expression of dystrophin 3’ mRNA in nuclei that appear to be outside the membrane boundary of the revertant fiber. The authors should provide a better example of the 3’ probe signal being evident within revertants.

With respect, the critical diagnostic feature of revertant fibres is the presence of small sarcoplasmic foci of both probes (5’ and 3’), which as shown are within the fibres identified as revertant in serial section. Presence of 3’ foci alone (without adjacent 5’) is consistent with shorter dystrophin isoforms (most likely dp71, for example in myoblasts or endothelia). We see no reason why the two expression patterns cannot be present within the same imaging field, and have added this clarifying detail to the figure legend.

3) Figure 9. Please provide a consecutive section stained for dystrophin protein to corroborate the RNA data, showing that with high levels of PPMO, dystrophin protein is detected. The animal was treated for many weeks and protein expression should be detectable.

As noted within the manuscript, these samples were taken from two separate studies: the 12.5mg.kg-1 data is public domain, and samples do indeed show substantial (widespread and uniform) dystrophin restoration. 

https://academic.oup.com/hmg/article/24/15/4225/2453018

We reference the manuscript in question and the numbers obtained therein, but we are happy to edit the text to make this referencing clearer. Regarding inclusion of these separate findings here: we do not wish to risk accusations of publishing the same data twice. With the editor’s approval we would however be happy to include data from this separate study, acknowledged as such clearly and appropriately.

4) The PPMO data is gathered from a single mouse, single muscle. It is felt that this seems to be a limited observation overall that should be further substantiated. At the very least, multiple muscles should be analyzed from the mouse to confirm the findings seen in the quadriceps.

Again, these samples (two mice: one treated at 6mg.kg-1, one at 12.5mg.kg-1) are taken from two separate studies: the levels of transcript skipping and dystrophin protein restoration are already known (and published), with N=6-7 per treatment. We have taken a single representative sample from each study to show that our ISH approach robustly detects mature transcripts where mature transcripts would be expected (12.5mg.kg-1) but does not where much lower numbers would be expected (6mg.kg-1). As we further show, ISH can reveal complementary spatial information that more conventional measurements (such as qPCR) would not, but given the expense of this ISH approach we expect most studies to continue employing conventional methods for first-pass efficacy assessment. 

We have edited the manuscript to acknowledge that this final data set represents a limited proof of principle observation, and made additional edits to clarify that we are not making an efficacy assessment, but are instead determining whether ISH can detect rescued dp427 transcripts. 

“This data is only proof of principle (and the expense of this technique likely limits widespread diagnostic utility), but it illustrates potential limitations in measuring skipping percentages via qPCR alone”

5) Similarly, in Figure 8 determinations were made by analyzing a total of 60-120 particles (page 23) per mouse, which seems rather low to draw robust conclusions.

The reviewer raises an excellent consideration: how many measurements constitutes ‘enough’? A consideration of the data would suggest that the small sarcoplasmic 5’ foci should be tractable to sample-size analysis: as these represent single dp427 transcripts, they should exhibit a mean fluorescence intensity consistent with this. 

Given the reviewer’s comment, we have applied post hoc sample size calculations (Snedecor and Cochran, statistical methods, 1982) to the data for these foci: with the standard deviations we measure, determining the true mean with a precision of +/-10%, at a confidence level of 95%, requires analysis of 70-80 foci. The numbers of small 5’ foci used for this analysis (for each animal) exceed these values. We are thus confident our measured values reflect the mean with sufficient accuracy to validate our comparative analysis. 

We thank the reviewer for raising this important question and have added this statistical analysis to the manuscript accordingly.

---

## [Decision Letter · Decision Letter 1]

8 Sep 2020

Multiplex *in situ* hybridization within a single transcript: RNAscope reveals dystrophin mRNA dynamics

PONE-D-20-09506R1

Dear Dr. Hildyard,

We’re pleased to inform you that your manuscript has been judged scientifically suitable for publication and will be formally accepted for publication once it meets all outstanding technical requirements.

Kind regards,

Atsushi Asakura, Ph.D

Academic Editor

PLOS ONE

Additional Editor Comments (optional):

Reviewers' comments:

Reviewer's Responses to Questions

**Comments to the Author**

1. If the authors have adequately addressed your comments raised in a previous round of review and you feel that this manuscript is now acceptable for publication, you may indicate that here to bypass the “Comments to the Author” section, enter your conflict of interest statement in the “Confidential to Editor” section, and submit your "Accept" recommendation.

Reviewer #1: All comments have been addressed

2. Is the manuscript technically sound, and do the data support the conclusions?

Reviewer #1: Yes

3. Has the statistical analysis been performed appropriately and rigorously? 

Reviewer #1: Yes

4. Have the authors made all data underlying the findings in their manuscript fully available?

Reviewer #1: Yes

5. Is the manuscript presented in an intelligible fashion and written in standard English?

Reviewer #1: Yes

6. Review Comments to the Author

Reviewer #1: (No Response)

7. PLOS authors have the option to publish the peer review history of their article (what does this mean?). If published, this will include your full peer review and any attached files.

Reviewer #1: No

---

## [Editor Report · Acceptance letter]

15 Sep 2020

PONE-D-20-09506R1 

Multiplex *in situ* hybridization within a single transcript: RNAscope reveals dystrophin mRNA dynamics 

Dear Dr. Hildyard:

I'm pleased to inform you that your manuscript has been deemed suitable for publication in PLOS ONE. Congratulations! Your manuscript is now with our production department. 

Kind regards, 

on behalf of

Dr. Atsushi Asakura 

Academic Editor

PLOS ONE